# Heritability estimates for 361 blood metabolites across 40 genome-wide association studies

Fiona A. Hagenbeek[1,2]*, René Pool [1,2], Jenny van Dongen [1,2], Harmen H.M. Draisma[1], Jouke Jan Hottenga[1], Gonneke Willemsen[1], Abdel Abdellaoui[1], Iryna O. Fedko[1], Anouk den Braber [1,3,4], Pieter Jelle Visser [3,5], Eco J.C.N. de Geus [1,2,4], Ko Willems van Dijk [6,7,8], Aswin Verhoeven [9], H. Eka Suchiman [10], Marian Beekman [10], P. Eline Slagboom [10], Cornelia M. van Duijn[11], BBMRI Metabolomics Consortium, Amy C. Harms [12], Thomas Hankemeier[12], Meike Bartels [1,2,4], Michel G. Nivard [1,2,4,51]* & Dorret I. Boomsma [1,2,4,51]*

Metabolomics examines the small molecules involved in cellular metabolism. Approximately 50% of total phenotypic differences in metabolite levels is due to genetic variance, but heritability estimates differ across metabolite classes. We perform a review of all genome-wide association and (exome-) sequencing studies published between November 2008 and October 2018, and identify >800 class-specific metabolite loci associated with metabolite levels. In a twin-family cohort ($N = 5117$), these metabolite loci are leveraged to simultaneously estimate total heritability ($h^2_{total}$), and the proportion of heritability captured by known metabolite loci ($h^2_{Metabolite-hits}$) for 309 lipids and 52 organic acids. Our study reveals significant differences in $h^2_{Metabolite-hits}$ among different classes of lipids and organic acids. Furthermore, phosphatidylcholines with a high degree of unsaturation have higher $h^2_{Metabolite-hits}$ estimates than phosphatidylcholines with low degrees of unsaturation. This study highlights the importance of common genetic variants for metabolite levels, and elucidates the genetic architecture of metabolite classes.

[1] Department of Biological Psychology, Vrije Universiteit Amsterdam, Amsterdam, The Netherlands. [2] Amsterdam Public Health Research Institute, Amsterdam, The Netherlands. [3] Alzheimer Center Amsterdam, Department of Neurology, VU Amsterdam, Amsterdam UMC, Amsterdam, The Netherlands. [4] Amsterdam Neuroscience, Amsterdam, The Netherlands. [5] Department of Psychiatry and Neuropsychology, School of Mental Health and Neuroscience, Alzheimer Center Limburg, Maastricht University, Maastricht, The Netherlands. [6] Einthoven Laboratory for Experimental Vascular Medicine, Leiden University Medical Center, Leiden, The Netherlands. [7] Department of Human Genetics, Leiden University Medical Center, Leiden, The Netherlands. [8] Department of Internal Medicine, Division of Endocrinology, Leiden University Medical Center, Leiden, The Netherlands. [9] Center for Proteomics and Metabolomics, Leiden University Medical Center, Leiden, The Netherlands. [10] Department of Biomedical Data Sciences, Section of Molecular Epidemiology, Leiden University Medical Center, Leiden, The Netherlands. [11] Department of Epidemiology, Erasmus Medical Center, Rotterdam, The Netherlands. [12] Division of Analytical Biosciences, Leiden Academic Center for Drug Research, Leiden University and The Netherlands Metabolomics Centre, Leiden, The Netherlands. [51]These authors contributed equally: Michel G. Nivard, Dorret I. Boomsma. A full list of consortium members appears at the end of the paper. *email: f.a.hagenbeek@vu.nl; m.g.nivard@vu.nl; di.boomsma@vu.nl

The metabolome is defined as the collection of metabolites, i.e., small molecules involved in cellular metabolism, which are produced in cells[1] and can be categorized into many classes[2]. The overall aim of the field of metabolomics is to provide a holistic overview of the metabolome[1], and its role in biological mechanisms and metabolic disturbances in diseases. Elucidating this role may offer new therapeutic targets or new biomarkers for disease diagnosis[3]. Variation in metabolite levels can arise due to gender[4], and age[5], as well as physiologic effects, behavior, and lifestyle factors, such as diet[6]. Genetic differences may be a source of direct variation in metabolomics profiles, or an indirect source of variation through genetic influences on physiology, behavior, and (or) lifestyle.

Genome- and metabolome-wide analysis of common genetic variants in human metabolism have successfully identified genetically influenced metabolites[7]. In 2008, the first genome-wide association study (GWAS; $N = 284$ participants) identified four genetic variants associated with metabolite levels[8]. Thereafter, GWAS with increasing sample sizes, and in diverse populations, identified hundreds of single nucleotide polymorphism (SNP) associations with metabolites from a wide range of metabolite classes[7]. Additional metabolite loci have been identified by leveraging low-frequency and rare-variant analyses using (exome-) sequencing. We conducted a comprehensive review of all quantitative trait loci (QTL) discovery for metabolites and supply the complete reference list in Supplementary Table 1.

Twin and family studies have established that the heritability ($h^2$; proportion of phenotypic variance due to genetic factors) of metabolite levels is 50% on average, with a range from $h^2 = 0\%$ to $h^2 = 80\%$[6,9–16]. Several studies reported differences in heritability estimates among different classes of lipid species[13,15] or lipoprotein subclasses[14]. For example, Rhee et al.[12] reported higher heritability estimates for amino acids than for lipids. Essential amino acids, which cannot be synthesized by an organism de novo[17], had lower heritability than nonessential amino acids[12], that are synthesized within the body[17]. Several techniques are available to estimate the contribution of measured SNPs to trait heritability[18], and, given SNP data in family members, to simultaneously estimate SNP-associated ($h^2_{SNP}$) and pedigree-associated genetic variance ($h^2_{ped}$)[19]. Together the SNP- and pedigree-associated genetic effects account for the narrow-sense heritability. However, when including data of family members, the variance explained by genetic effects ($h^2_{total}$) may be biased upwards by shared environmental factors and/or nonadditive genetic effects[19,20].

An improved understanding of the genetic background of the metabolome will benefit our understanding of the etiology of diseases and traits, such as cardiometabolic diseases[21], migraine[22], psychiatric disorders[23], and cognition[24]. Here, we aim to further our understanding of the contribution of genetic factors to variation in fasting blood metabolic measures (henceforth referred to as metabolites for brevity) by the analysis of data from multiple metabolomics platforms in a large cohort of twins and family members ($N = 5117$). Specifically, we aim to estimate the total genetic variance of metabolite levels ($h^2_{total}$), and to elucidate the contribution to metabolite levels of known metabolite class-specific and metabolite class-unspecific loci ($h^2_{Metabolite-hits}$), on the basis of the results of a decade of GWA and (exome-) sequencing studies. To this end, we characterize all metabolite-SNP associations published between November 2008 and October 2018 by metabolite classification, and used linear-mixed models to estimate the $h^2_{total}$, $h^2_{SNP}$, and $h^2_{Metabolite-hits}$ simultaneously for 369 metabolites. In these models, the $h^2_{Metabolite-hits}$ consists of two variance components, a component attributable to metabolite loci associated with metabolites of a specific superclass ($h^2_{Class-hits}$) and a component attributable other metabolite loci ($h^2_{Notclass-hits}$).

The median $h^2_{total}$ for lipids is 0.47 and for organic acids 0.40, and the median lipid $h^2_{Metabolite-hits}$ is 0.06 and 0.01 for organic acids, with most of the $h^2_{Metabolite-hits}$ attributable to $h^2_{Class-hits}$. We further expand on the current knowledge of the genetic etiology of metabolite classes by employing mixed-effect meta-regression models to test differences in heritability estimates among metabolite classes and among lipid species. Although estimates of $h^2_{total}$ do not differ significantly among metabolite classes, significant differences were observed among lipid and organic classes for $h^2_{Metabolite-hits}$ and $h^2_{Class-hits}$.

Intriguingly, phosphatidylcholines[11] and triglycerides (TGs)[16] show increasing heritability with increasing number of carbon atoms and/or double bonds in their fatty acyl side chains. Draisma et al.[11] speculated this might be attributable to differences in the number of metabolic conversion rounds for phosphatidylcholines or TGs with a variable number of carbon atoms. To distinguish between the effects of the number of carbon atoms or number of double bonds in the fatty acyl side chains of phosphatidylcholines and TGs, we conduct additional univariate follow-up analyses. Our results indicate higher $h^2_{Metabolite-hits}$ estimates for more complex phosphatidylcholines (i.e., with larger number of carbon atoms and/or double bonds). Univariate follow-up suggests this could be attributed to the number of double bonds in phosphatidylcholines (e.g., degree of unsaturation).

## Results

**Metabolite classification.** In the period of November 2008 to October 2018, 40 GWA and (exome-) sequencing studies identified 242,580 metabolite-SNP or metabolite ratio-SNP associations (see Supplementary Table 1). All 242,580 associations may be found in Supplementary Data 1, which lists the significant SNP-metabolite associations by study. These associations, included 1804 unique metabolites or ratios and 49,231 unique SNPs (43,830 after converting all SNPs to NCBI build 37; Supplementary Data 1). The human metabolome database (HMDB)[2] identifiers of each metabolite were retrieved in order to extract information concerning the metabolite's hydrophobicity and chemical classification (see Methods). Excluding the ratios and unidentified metabolites, we classified 953 metabolites into 12 super classes (Table 1), 43 classes, or 77 subclasses based on the HMDB classification (Supplementary Data 1). The majority of the metabolites were classified into the super classes lipids or organic acids. The lipids

**Table 1 Overview of the number of unique metabolites per super class.**

| Super class | Number of unique metabolites |
|---|---|
| Lipids and lipid-like molecules (e.g., lipids) | 662 |
| Organic acids and derivatives (e.g., organic acids) | 182 |
| Organoheterocyclic compounds | 45 |
| Organic oxygen compounds | 19 |
| Nucleosides, nucleotides, and analogues | 12 |
| Benzenoids | 12 |
| Organic nitrogen compounds | 11 |
| Phenylpropanoids and polyketides | 4 |
| Proteins | 3 |
| Organic compounds | 1 |
| Trichlorophenols | 1 |
| Organooxygen compounds | 1 |

For each Human Metabolome[2] super class the number of unique metabolites, for which significant SNP-metabolite associations have been published, is provided. See Supplementary Data 1 for an overview of the exact metabolites classified per super class, class, and subclass, as well as the SNPs associated with each metabolite

could be subdivided into 8 classes, with 1 to 95,795 metabolite-SNP associations per class (mean = 17,589; SD = 32,553), and in 32 subclasses, with the number of subclass metabolites-SNP associations ranging from 1 to 40,440 (mean = 4673; SD = 9124). The organic acids and derivatives were divided in 9 classes, with the number of metabolite-SNP associations ranging from 1 to 26,832 (mean = 3374; SD = 8832). The organic acids and derivatives were also divided into 17 organic acid subclasses, with the number of subclass metabolite-SNP associations ranging from 1 to 26,448 (mean = 1786; SD = 6371; Supplementary Data 1). Across all four platforms 427 metabolites were assessed. After excluding the ratios (17) and the metabolites of super classes not included in the curated metabolite-SNP association list (8), data were available for 402 metabolites. The full list of metabolites, with their classifications and the quartile values of the untransformed levels, is included in Supplementary Table 1. The 402 metabolites were classified as 336 lipids, 53 organic acids, 9 organic oxygen compounds, 3 proteins and one organic nitrogen compound. These super classes consisted of 12 classes (Supplementary Table 2). In this paper we mainly focus on the first two super classes. After quality control (QC), 369 metabolites from these two super classes were retained for analysis.

**Characterization of the heritable influences on metabolites.** Data of 5117 participants were available from the following four metabolomics platforms: the Nightingale Health proton nuclear magnetic resonance ($^1$H-NMR) platform, a ultra performance liquid chromatography mass spectrometry (UPLC-MS) lipidomics platform, the Leiden $^1$H-NMR platform, and the Biocrates Absolute-IDQ$^{TM}$ p150 platform. The participants were registered with the Netherlands Twin Register (NTR)[25] and were clustered in 2445 nuclear families. Metabolomics and SNP data were available for all participants. Background and demographic characteristics for the sample can be found in Table 2.

We aimed to assess the variance explained by previously identified metabolite GWA and (exome-) sequencing genetic variants in our (independent) sample. Clearly, our results are conditional on the power of past the studies, as the list of metabolite genetic variants is based on previous GWA and (exome-) sequencing studies, which vary in power. We present the sample size of each past study in Supplementary Table 1, and the sample size per metabolite-SNP association in Supplementary Data 1.

Linear-mixed models including all loci for genetic variants associated with metabolites in a single genetic relatedness matrix (GRM) will contain SNPs that are associated with some metabolites, but not with others, or include many SNPs that are not associated with a given metabolite. We therefore created two GRMs for the loci associated with metabolite hits (see Methods): one class-specific and one nonclass specific (i.e., GRMs including metabolite loci for all metabolites, except for the target metabolite class). We explored models for the 12

class-specific and the corresponding not-class specific GRMs (Supplementary Note 2). These models displayed high degrees of non-convergence (37.9% total), with models including small class-specific GRMs displaying more non-convergence (Supplementary Table 2). Therefore, the results in the remainder of this paper were based on the metabolite super classes, i.e., lipids and organic acids.

For the 369 lipids and organic acids, we carried out unconstrained four-variance component analyses (Fig. 1). In genome-wide complex trait analysis (GCTA)[18] we specified a model in which we partition the metabolite variation into SNP-associated ($h^2_{SNP}$), pedigree-associated ($h^2_{ped}$), class-specific metabolite-loci-associated ($h^2_{class-hits}$), and not-class metabolite-loci-associated ($h^2_{notclass-hits}$) genetic variation (Fig. 1). We report the total heritability ($h^2_{total}$), the proportion attributable to metabolite superclass-specific loci ($h^2_{Class-hits}$), the proportion of variance attributable to non-superclass metabolite loci ($h^2_{Notclass-hits}$) and the contribution of known metabolite loci to metabolite levels ($h^2_{Metabolite-hits}$). The analyses were performed separately for lipids and organic acids, with class-specific and corresponding nonclass GRMs (created using the LDAK program[26,27]) in both sets of analyses. The lipid analyses employed a class-specific GRM of 479 lipid loci and a corresponding nonclass GRM of 596 loci (Supplementary Fig. 1). The organic acid analyses included a class-specific GRM of 397 loci and a nonclass GRM of 683 loci (Supplementary Fig. 1). Before the analyses, the metabolite data were normalized (log-normal or inverse rank; see Methods). All models included age at blood draw, sex, the first ten principal

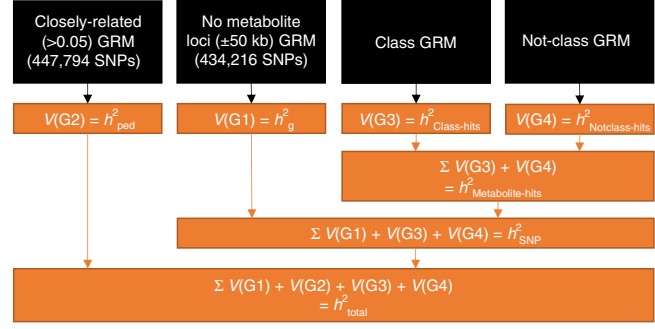

**Fig. 1 Overview of the four-variance component models.** Overview of the SNP-filtering and GRM construction can be found in Supplementary Fig. 1 and is explained in details in the Methods. This figure describes which GRMs (black boxes) are used to calculate which variance components (orange boxes) by drawing black arrows from the GRMs to the variance components. The variance components give rise to the four different heritability estimates: $h^2_{ped}$, $h^2_g$, $h^2_{Class-hits}$, and $h^2_{Notclass-hits}$ (see Methods). The orange arrows indicate how the various variance components are summed to obtain estimates for $h^2_{metabolite-hits}$, $h^2_{SNP}$, and $h^2_{total}$ (see Methods).

| Table 2 Participant characteristics per metabolomics platform. | | | | | | | | | |
|---|---|---|---|---|---|---|---|---|---|
| Metabolomics platform | N | N families | Age[a] (mean ± SD) | Female (%) | Twins (%) | BMI (mean ± SD) | Cholesterol[b] (mean ± SD) | LDL[b] (mean ± SD) | HDL[b] (mean ± SD) |
| All participants | 5117 | 2445 | 42.1 ± 14.2 | 62.8 | 63.4 | 24.8 ± 4.1 | 4.9 ± 1.2 | 3.0 ± 1.0 | 1.7 ± 1.0 |
| Nightingale Health $^1$H-NMR | 4227 | 2179 | 40.7 ± 13.7 | 67.3 | 69.7 | 24.6 ± 4.0 | 4.9 ± 1.2 | 3.0 ± 1.0 | 1.7 ± 1.0 |
| UPLC-MS lipidomics | 2324 | 1251 | 39.0 ± 12.9 | 66.6 | 89.2 | 24.4 ± 4.1 | 5.0 ± 1.0 | 3.0 ± 0.9 | 1.4 ± 0.4 |
| Leiden $^1$H-NMR | 2324 | 1323 | 37.6 ± 12.5 | 67.0 | 89.0 | 24.2 ± 4.1 | 4.6 ± 1.3 | 2.7 ± 1.0 | 2.0 ± 1.4 |
| Biocrates | 1448 | 946 | 45.7 ± 15.3 | 43.8 | 39.6 | 25.2 ± 3.9 | 4.6 ± 1.5 | 2.8 ± 1.1 | 2.3 ± 1.7 |

This table gives an overview of the number of individuals (N) per platform, specifies the number of families these individuals belong to and the percentage of females and twins in each dataset. In addition, for each platform the mean and standard deviation (SD) of the age at blood draw in years, the body mass index (BMI), the cholesterol level in mmol/l, the low-density lipoprotein cholesterol (LDL) levels in mmol/l, and the high-density lipoprotein cholesterol (HDL) levels in mmol/l are given. All participant characteristics are given after preprocessing, which was done separately for each metabolomics platform (see Methods)
[a]Age at blood draw in years
[b]Levels in mmol/l

**Table 3 Summary of the heritability estimates of the four-variance component models.**

| | | Mean | Median | Range |
|---|---|---|---|---|
| Lipids and lipid-like molecules | $h^2_{\text{total}}$ estimate | 0.47 | 0.47 | (0.11–0.66) |
| | $h^2_{\text{total}}$ s.e. | 0.04 | 0.03 | (0.02–0.07) |
| | $h^2_{\text{Metabolite-hits}}$ estimate | 0.06 | 0.06 | (−0.05–0.16) |
| | $h^2_{\text{Metabolite-hits}}$ s.e. | 0.03 | 0.03 | (0.01–0.04) |
| | $h^2_{\text{Class-hits}}$ estimate | 0.06 | 0.06 | (−0.02–0.16) |
| | $h^2_{\text{Class-hits}}$ s.e. | 0.02 | 0.02 | (0.01–0.03) |
| | $h^2_{\text{Notclass-hits}}$ estimate | 0.00 | 0.01 | (−0.06–0.12) |
| | $h^2_{\text{Notclass-hits}}$ s.e. | 0.02 | 0.02 | (0.01–0.03) |
| Organic acids and derivatives | $h^2_{\text{total}}$ estimate | 0.41 | 0.40 | (0.14–0.72) |
| | $h^2_{\text{total}}$ s.e. | 0.04 | 0.03 | (0.02–0.07) |
| | $h^2_{\text{Metabolite-hits}}$ estimate | 0.01 | 0.02 | (−0.08–0.11) |
| | $h^2_{\text{Metabolite-hits}}$ s.e. | 0.02 | 0.02 | (0.01–0.04) |
| | $h^2_{\text{Class-hits}}$ estimate | 0.01 | 0.01 | (−0.04–0.14) |
| | $h^2_{\text{Class-hits}}$ s.e. | 0.02 | 0.02 | (0.01–0.03) |
| | $h^2_{\text{Notclass-hits}}$ estimate | 0.00 | 0.00 | (−0.06–0.05) |
| | $h^2_{\text{Notclass-hits}}$ s.e. | 0.02 | 0.02 | (0.01–0.03) |

The mean, median, and range of the total heritability ($h^2_{\text{total}}$), heritability based on the 479 significant metabolite loci for the 309 lipids or the 397 significant metabolite loci for the 52 organic acids ($h^2_{\text{Class-hits}}$), the 596–683 significant metabolite loci not belonging to these classes ($h^2_{\text{Notclass-hits}}$) and the total heritability explained by metabolite loci (e.g., sum of $h^2_{\text{Class-hits}}$ and $h^2_{\text{Notclass-hits}}$: $h^2_{\text{Metabolite-hits}}$), as well as their standard errors (s.e.'s), are depicted for all 361 successfully analyzed metabolites as included on all platforms. Supplementary Data 2 denotes which metabolites belong to each class and Supplementary Data 3 provides the estimates for each of the individual metabolites

components (PCs) from SNP genotype data, genotyping chip, and metabolomics measurement batch as covariates.

Supplementary Data 3 includes the estimates from the four-variance genetic component models for all 369 metabolites. The genomic relatedness matrix residual maximum likelihood (GREML) algorithm converged for 361 (97.8%) of the 53 organic acids and 316 lipids (Supplementary Table 3). Non-convergence of the GREML algorithm was observed for 6 metabolites (1.6%). The analyses of 2 metabolites (0.5%) were not completed due to non-invertible variance-covariance matrices. The estimates for $h^2_{\text{total}}$ of the 309 lipids ranged from 0.11 to 0.66 (mean = 0.47; mean s.e. = 0.04). The estimates for $h^2_{\text{Metabolite-hits}}$ ranged from −0.05 to 0.16 (mean = 0.06; mean s.e. = 0.03; Table 3). The 52 organic acids had $h^2_{\text{total}}$ estimates ranging from 0.14 to 0.72 (mean = 0.41; mean s.e. = 0.04). The estimates for $h^2_{\text{Metabolite-hits}}$ ranged from −0.08 to 0.11 (mean = 0.01; mean s.e. = 0.02; Table 3). On average, for both lipids and organic acids the $h^2_{\text{class}}$ was higher than the $h^2_{\text{Notclass}}$, with $h^2_{\text{Class-hits}}$ ranging from −0.02 to 0.16 (0.06; mean s.e. = 0.02) for lipids and from −0.04 to 0.14 for organic acids (mean = 0.01; mean s.e. = 0.02). For both lipids and organic acids $h^2_{\text{Notclass-hits}}$ was zero (mean s.e. = 0.02), ranging from −0.06 to 0.12 for lipids and from −0.06 to 0.05 for organic acids (Table 3).

Including multiple metabolomics platforms allowed for a comparison of metabolites as measured on multiple platforms. An earlier study showed that 29 out of 43 metabolites present on two platforms to exhibit moderate heritability on both platforms[28]. In the current study, 61 metabolites were measured on multiple platforms (phenotypic correlations provided in Supplementary Data 4), with moderate $h^2_{\text{total}}$ on each of the platforms and on average a positive correlation of 0.36 between the $h^2_{\text{total}}$ of the same metabolite assessed on different platforms (Supplementary Data 4).

**Differential heritability among metabolite classes**. Figure 2 shows variation in median heritability among the following classes of organic acids: keto acids, hydroxy acids, and carboxylic acids (see Supplementary Data 2 for metabolites per class). Keto acids, followed by carboxylic acids, had the highest median $h^2_{\text{total}}$, and $h^2_{\text{Class-hits}}$ estimates (Fig. 2). While hydroxy acids had the highest median $h^2_{\text{Notclass-hits}}$ and $h^2_{\text{Metabolite-hits}}$ estimates, the lowest median $h^2_{\text{total}}$, and $h^2_{\text{Class-hits}}$ estimates were observed for these metabolites

(Fig. 2). To investigate whether heritability differs significantly among classes of organic acids, we applied multivariate mixed-effect meta-regression, corrected for metabolite platform effects (see Methods). The multivariate mixed-effect meta-regression models showed that $h^2_{\text{total}}$ and $h^2_{\text{Class-hits}}$ for the organic acid classes did not differ significantly. However, significant differences among the organic acid classes were observed with multivariate mixed-effect meta-regression models with respect to the $h^2_{\text{Metabolite-hits}}$ estimates ($F(4, 47) = 3.44$, false discovery rate (FDR)-adjusted $p$ value = 0.03), and the $h^2_{\text{Notclass-hits}}$ estimates ($F(4, 47) = 19.95$, FDR-adjusted $p$ value = $1.25 \times 10^{-08}$; Supplementary Data 5).

The multivariate mixed-effect meta-regressions were also applied to assess the significance of heritability differences among essential and non-essential amino acids (subdivision of carboxylic acids; see Supplementary Table 4) and among lipid classes (see Supplementary Data 2 for metabolites per lipid class). The meta-regression analyses revealed no significant mean differences among essential and non-essential amino acids (Table 4; Supplementary Data 6). Small but significant mean heritability differences were observed with multivariate mixed-effect meta-regression models among the different classes of lipids (Fig. 3). For lipid classes the $h^2_{\text{Metabolite-hits}}$ estimates differed significantly ($F(8, 300) = 8.47$; FDR-adjusted $p$ value = 0.004; Supplementary Data 5).

Finally, we explored whether heritability of phosphatidylcholines and TGs increases with a larger number of carbon atoms and/or double bonds in their fatty acyl side chains. To this end we employed both uni- and multivariate mixed-effect meta-regression models separately for the TGs, diacyl phosphatidylcholines (PCaa) and acyl-alkyl phosphatidylcholines (PCae; see Methods). The platform specific heritability estimates for each of these lipid species are depicted in Supplementary Fig. 2. Multivariate mixed-effect meta-regression models showed that variation in the number of carbon atoms and double bonds was significantly associated with $h^2_{\text{Metabolite-hits}}$ estimates for PCaa's ($F(3, 52) = 7.05$; FDR-adjusted $p$ value = 0.009) and PCae's ($F(3, 45) = 3.41$; FDR-adjusted $p$ value = 0.05; Supplementary Data 5). Phosphatidylcholines with a larger number of carbon atoms showed lower heritability estimates and phosphatidylcholines with a larger number of double bonds had higher heritability estimates (Supplementary Data 5). The differences among the phosphatidylcholines with a variable number of carbon atoms and/or double

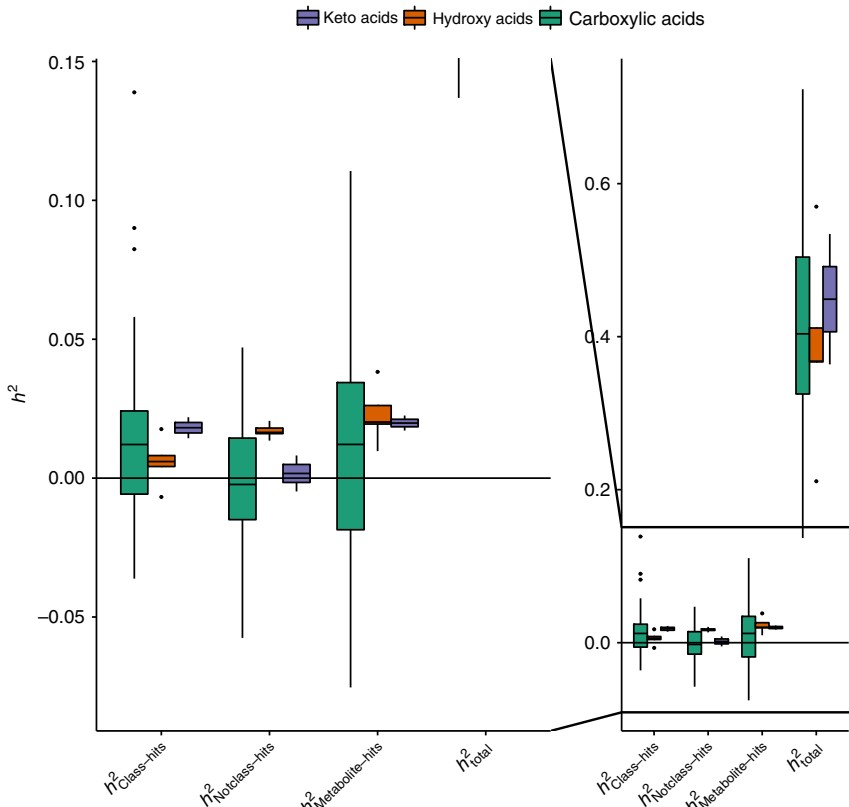

**Fig. 2 Heritability of all 52 carboxylic acids by class.** Box- and dotplots of the $h^2_{total}$, and $h^2_{Metabolite-hits}$ for all 52 successfully analyzed carboxylic acids and derivatives across all metabolomics platforms by class. The left-hand side of the figure is a close-up of the $-0.08$ to 0.15 part of the heritability range, focusing on the $h^2_{Class-hits}$ and $h^2_{Notclass-hits}$ estimates. The boxes denote the 25th and 75th percentile (bottom and top of box), and median value (horizontal band inside box). The whiskers indicate the values observed within up to 1.5 times the interquartile range above and below the box. The purple, orange and green boxes denote the keto acid, hydroxyl acid and carboxylic acid classes, respectively. Supplementary Data 3 provides the estimates for each of the individual metabolites.

**Table 4 Summary of the heritability estimates for the essential and nonessential amino acids.**

|  |  | Mean | Median | Range |
|---|---|---|---|---|
| Essential amino acids | $h^2_{total}$ estimate | 0.42 | 0.40 | (0.23–0.64) |
|  | $h^2_{total}$ s.e. | 0.04 | 0.03 | (0.02–0.07) |
|  | $h^2_{Metabolite-hits}$ estimate | 0.00 | 0.00 | (−0.05–0.05) |
|  | $h^2_{Metabolite-hits}$ s.e. | 0.02 | 0.02 | (0.01–0.03) |
|  | $h^2_{Class-hits}$ estimate | 0.01 | 0.00 | (−0.03–0.05) |
|  | $h^2_{Class-hits}$ s.e. | 0.02 | 0.02 | (0.01–0.02) |
|  | $h^2_{Notclass-hits}$ estimate | −0.01 | −0.01 | (−0.06–0.04) |
|  | $h^2_{Notclass-hits}$ s.e. | 0.02 | 0.02 | (0.01–0.03) |
| Non-essential amino acids | $h^2_{total}$ estimate | 0.39 | 0.39 | (0.22–0.69) |
|  | $h^2_{total}$ s.e. | 0.04 | 0.04 | (0.03–0.07) |
|  | $h^2_{Metabolite-hits}$ estimate | 0.02 | 0.01 | (−0.07–0.11) |
|  | $h^2_{Metabolite-hits}$ s.e. | 0.03 | 0.03 | (0.01–0.04) |
|  | $h^2_{Class-hits}$ estimate | 0.03 | 0.01 | (−0.03–0.14) |
|  | $h^2_{Class-hits}$ s.e. | 0.02 | 0.02 | (0.01–0.03) |
|  | $h^2_{Notclass-hits}$ estimate | 0.00 | 0.00 | (−0.04–0.03) |
|  | $h^2_{Notclass-hits}$ s.e. | 0.02 | 0.02 | (0.01–0.03) |

The mean, median, and range of the total heritability ($h^2_{total}$), and heritability based on the 397 significant metabolite loci for the organic acids ($h^2_{Class-hits}$), the 683 significant metabolite loci not belonging to this class ($h^2_{Notclass-hits}$) and the total heritability explained by metabolite loci (e.g., sum of $h^2_{Class-hits}$ and $h^2_{Notclass-hits}$: $h^2_{Metabolite-hits}$), as well as their standard errors (s.e.'s), are depicted for all 31 successfully analyzed essential (17) and nonessential (14) amino acids as included on all platforms. Supplementary Data 2 denotes which metabolites belong to each class and Supplementary Data 3 provides the estimates for each of the individual metabolites

bonds may have contributed to differential $h^2_{Class}$ estimates. Univariate models confirmed the results for the number of double bonds in PCaa's and PCae, though they were not significant after correction for multiple testing (Supplementary Data 6).

## Discussion

We carried out a comprehensive assessment of GWA-metabolomics studies, and created a repository of all studies reporting on associations of SNPs and blood metabolites in

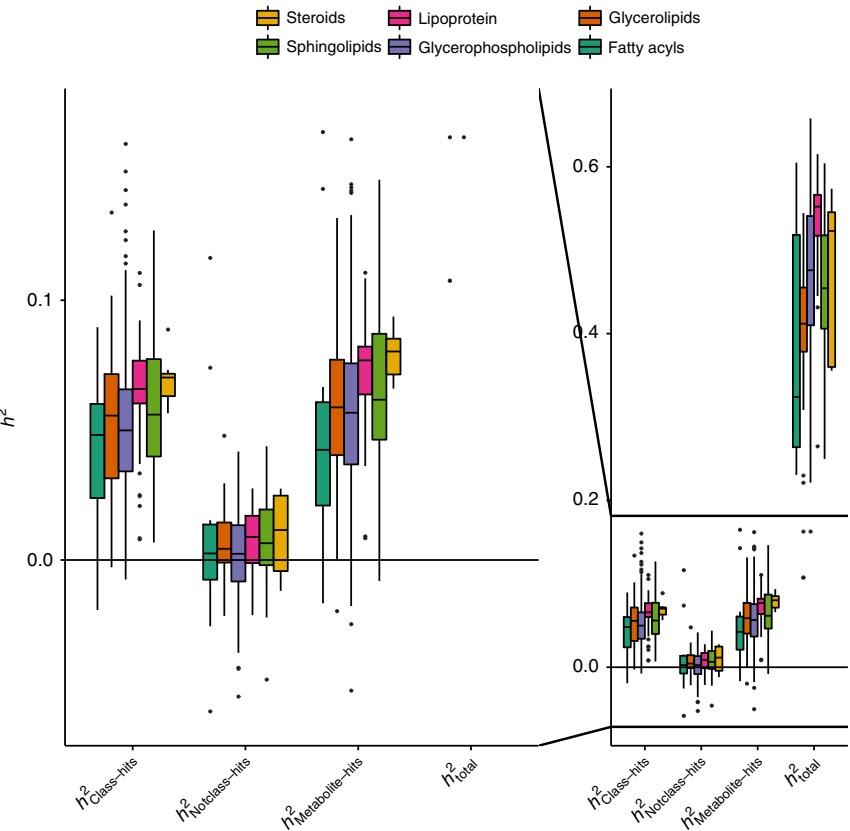

**Fig. 3 Heritability of all 309 lipids by class.** Box- and dotplots of the $h^2_{total}$, and $h^2_{Metabolite-hits}$ for all 309 successfully analyzed lipids and lipid-like molecules across all metabolomics platforms by class. The left-hand side of the figure is a close-up of the −0.06 to 0.17 part of the heritability range, focusing on the $h^2_{Class-hits}$ and $h^2_{Notclass-hits}$ estimates. The boxes denote the 25th and 75th percentile (bottom and top of box), and median value (horizontal band inside box). The whiskers indicate the values observed within up to 1.5 times the interquartile range above and below the box. The yellow, pink, orange, light green, purple, and dark green boxes denote the steroids, lipoprotein, glycerolipid, sphingolipid, glycerophospholipid, and fatty acyl classes, respectively. Supplementary Data 3 provides the estimates for each of the individual metabolites.

European ancestry samples. We curated 241,965 genome-wide metabolite associations and we classified the associated metabolites into super classes, classes and subclasses. The complete overview of all blood metabolite-SNP associations is provided in Supplementary Data 1, with the complete list of references in Supplementary Table 1. The information from the repository was used to construct GRMs, which served to identify genetic variance components in the analysis of 369 metabolites. The metabolite data in our study came from a large cohort of twin-families ($N = 5117$ clustered in 2445 families) measured on four metabolomics platforms. We focused on two metabolite super classes. By mapping all metabolites to the HMDB[2] we were able to classify both the measured metabolites and all previously published metabolites as either lipids or organic acids. In the current study, we sought to elucidate the contribution of known metabolite loci, based on a decade of GWA and (exome-) sequencing studies, to metabolite levels ($h^2_{Metabolite-hits}$). A unique feature of our study was the ability to disentangle the role of class-specific ($h^2_{Class-hits}$) and nonclass ($h^2_{Notclass-hits}$) metabolite loci on heritability differences among metabolite classes and lipid species.

To evaluate differences among metabolite classes and lipid species in the estimates for $h^2_{total}$, we applied multivariate mixed-effect meta-regression models to the estimates of $h^2_{Metabolite-hits}$, $h^2_{Class-hits}$, and $h^2_{Notclass-hits}$. We observed no significant differences in $h^2_{total}$ estimates among the metabolite classes. Consistent with a previous twin-family study[10], none of the heritability estimates differed significantly among essential and nonessential amino acids. We observed significant $h^2_{Metabolite-hits}$ differences among the different classes of organic acids. Keto acids had significantly lower $h^2_{Metabolite-hits}$ estimates as compared with carboxylic acids. Class-specific metabolite loci heritability estimates for fatty acyls, lipoproteins and steroids were significantly higher. Similarly, significant heterogeneity in lipid class heritability, with lower $h^2_{total}$ and $h^2_{SNP}$ for phospholipids than for sphingolipids or glycerolipids has been reported[13,15,29]. Lastly, we assessed whether heritability increases with added complexity in lipid species[11,16]. We found that this was the case with respect to $h^2_{Metabolite-hits}$ estimates in more complex diacyl and acyl–alkyl phosphatidylcholines, but not for more complex TGs. Previous research reported significant higher $h^2_{SNP}$ estimates in poly-unsaturated fatty acid containing lipids[15]. Furthermore, loci associated with traditional lipid measures explained 2–21% of the variance in lipid levels[15]. Together these results suggest that higher heritability in phosphatidylcholines is driven by a lower number of carbon atoms and higher number of double bonds, e.g., a larger degree of unsaturation.

Evaluating the mean heritability differences among lipids and organic acids, it appears that lipids have higher $h^2_{total}$, $h^2_{Class-hits}$, and $h^2_{Metabolite-hits}$ estimates than organic acids (Table 3). Previous twin-family studies indicates that the heritability difference among lipids and organic acid is rarely investigated[9–12]. This is possibly because most metabolomics platforms focus mainly on either lipids or organic acids. Lipid metabolite classes tend to be very well represented on metabolomics platforms, whereas

organic acids are unrepresented, and as a consequence, the analysis to obtain $h^2_{Class-hits}$ and $h^2_{Metabolite-hits}$ estimates of the organic acids will be underpowered due to this imbalance.

The current study has several limitations. First, the extent to which our findings generalize to populations of non-European ancestry is unknown. Loci of common human metabolism pathways are most likely to replicate over ethnicities[30]. Second, estimates of the total variance explained may show upward bias when based on data from closely related individuals (e.g., first cousins or closer)[19,20]. This bias is caused by the influence of shared environmental influences, epistatic interactions, or dominance[19,20]. While the results of the current study may suffer of such biases by the inclusion of twins, siblings, and parents, the sample also includes many unrelated individuals which will reduce the possible bias (Supplementary Fig. 3).

Kettunen et al.[31] investigated 217 metabolites of the Nightingale Health [1]H-NMR platform in a classical twin design and reported dominance effects for 6.45% of the metabolites. Tsepsilov et al.[32] performed GWA study targeting nonadditive genetic effects and concluded that most genetic effects on metabolite levels and ratios were in fact additive. Together, these studies suggested that the bias due to dominance effects on metabolite levels will be minor.

Relatively few twin-family studies explicitly investigated the role of shared environmental influences on metabolite levels. Overall, shared environmental influences are reported for a small number of metabolites (e.g., 14.3% of all Nightingale Health [1]H-NMR metabolites[31]) and the influence of the shared environment is small-to-moderate (platform and metabolite class-dependent averages range from 0.03 to 0.45[6,13,33–35] with larger estimates deriving from small studies). For studies including parents and offspring, or adult twin and siblings pairs the question arises which effects are captured by the shared environment. Are these the lasting influences of the environment offspring shared with their parents and with each other before they started living independently? Additional research is necessary to elucidate the role of the shared environment on metabolite levels[19].

Third, standard errors of $h^2_{SNP}$ estimates were high. While we have included all $h^2_{SNP}$ estimates in the supplements, we stress that the primary goal of our paper was to investigate the contribution of known metabolite loci in an independent sample rather than obtaining the $h^2_{SNP}$ estimates for metabolites.

Finally, the estimates for $h^2_{metabolite-hits}$ are based on SNPs of 40 different studies from a decade of GWA and (exome-) sequencing studies. The sample size, and therefore the power, of these studies vary, with some studies conducted with as few as 211 individuals while others included over 24,000 individuals (Supplementary Table 1). For underrepresented metabolites the low power may result in downward biased heritability estimates. However, leveraging information from a decade of research in 40 studies and extracting loci for metabolite classes across multiple studies, the number of such metabolites is not large. New[29,36–38] and future studies will increase the number of variants identified as metabolite loci. The investment in UK Biobank[39] is expected to dramatically increase sample sizes for large-scale genomic investigations of the human metabolome and subsequently the number of metabolite loci.

Mendelian randomization may benefit from the comprehensive overview of metabolite loci that we identified. The identified loci can serve as instruments in metabolome-wide Mendelian randomization studies of complex traits. In addition, our work offers valuable insights into the role of common genetic variants in class specific heritability differences among metabolite classes and lipids species. Further research is required to elucidate the contribution of rare genetic variants to metabolite levels, and differences in the contribution of rare genetic variants among

metabolite classes. A reasonable approach would be to carry out a similar study in a large sample of whole-genome sequencing data. Such an approach, using minor allele frequency (MAF)- and linkage disequilibrium (LD)-stratified GREML analysis[40], identified additional variance due to rare variants for height and body mass index[41].

In conclusion, we contributed to our understanding of the genetic architecture of fasting blood metabolite levels, and of differences in the genetic architecture among metabolite classes. Extending the GREML framework with the inclusion of known metabolite loci allowed us to simultaneously estimate $h^2_{total}$, and $h^2_{metabolite-hits}$ (which consists of $h^2_{Class-hits}$ and $h^2_{Notclass-hits}$) for 361 metabolites. Significant differences in $h^2_{Metabolite-hits}$ estimates were observed among different classes of lipids and organic acids and for more complex diacyl and acyl–alkyl phosphatidylcholines. Future studies should address the proportion of metabolite variation influenced by heritable and nonheritable lifestyle factors, as this will facilitate the development of personalized disease prevention and treatment of complex disorders.

## Methods

**Participants.** At the NTR[42] metabolomics data for twins and family members as measured in blood samples were available for 6011 individuals of whom 5667 were genotyped. The blood samples for the four metabolomics experiments described in this study were mainly collected in participants of the NTR biobank project[25,43]. Blood samples were collected after a minimum of two hours of fasting (1.3%), with the majority of the samples collected after overnight fasting (98.7%). Fertile women were bled in their pill-free week or on day 2–4 of their menstrual cycle. For the current paper, we excluded participants who were not of European ancestry, who were on lipid-lowering medication at the time of blood draw, and who failed to adhere to the fasting protocol. The exact number of exclusions per dataset is listed in Supplementary Data 7. After completing the preprocessing of the metabolomics data, the separate subsets (e.g., different collection and measurement waves; see Supplementary Data 7) of each platform were merged into a single per platform dataset, retaining a single (randomly chosen) observation per platform when multiple observations were available. Supplementary Data 8 gives an overview of the overlap in participants among the different platforms, with the overlap among each metabolite that survived QC for all four platforms available in Supplementary Data 9. The final number of participants included in the study was 5117, with platform specific sample size ranging from 1448 to 4227 individuals clustered in 946–2179 families. Characteristics for the individuals can be found in Table 2. Supplementary Fig. 3 depicts the distribution of the relatedness in the sample. Informed consent was obtained from all participants. Projects were approved by the Central Ethics Committee on Research Involving Human Subjects of the VU University Medical Centre, Amsterdam, an Institutional Review Board certified by the U.S. Office of Human Research Protections (IRB number IRB00002991 under Federal-wide Assurance- FWA00017598; IRB/institute codes, NTR 03–180 and EMIF-AD 2014.210).

**Metabolite profiling.** Plasma and serum samples have been profiled on four metabolomics platforms: two proton nuclear magnetic resonance spectroscopy ([1]H-NMR) platforms and two mass spectrometry (MS) platforms. Plasma samples have been analyzed on the Nightingale Health [1]H-NMR platform (Nightingale Health Ltd., Helsinki, Finland), an MS lipidomics platform, and the Leiden [1]H-NMR platform. Serum samples were analyzed with the Biocrates Absolute-IDQ[TM] p150 platform (Biocrates Life Sciences AG, Innsbruck, Austria). Details about each of the metabolomics platforms have been included in Supplementary Note 2.

**Metabolomics data preprocessing.** Preprocessing of the metabolomics data was done separately for each of the platforms and each measurement batch. Metabolites were excluded from analysis when the mean coefficient of variation exceeded 25% and the missing rate exceeded 5%. Metabolite measurements were set to missing if they were below the lower limit of detection or quantification or could be classified as an outlier (five standard deviations greater or smaller than the mean). Metabolite measurements, which were set to missing because they fell below the limit of detection/quantification were imputed with half of the value of this limit, or when this limit was unknown with half of the lowest observed level for this metabolite. All remaining missing values were imputed using multivariate imputation by chained equations (mice)[44]. On average, 9 values were imputed for each metabolite (SD = 12; range: 1–151). Data for each metabolite on both [1]H-NMR platforms were normalized by inverse normal rank transformation[45,46], while the imputed values of the Biocrates metabolomics platform and the UPLC-MS lipidomics platform were normalized by natural logarithm transformation[11,47], conform previous normalization strategies applied to the data obtained using these platforms. The complete lists with full names of all detected metabolites that survived

QC and preprocessing for all platforms can be found in Supplementary Data 2, these tables also include the quartile values of the untransformed metabolites.

**Genotyping, imputation, and ancestry outlier detection.** Genotype information was available for 21,001 NTR participants from 6 different genotyping arrays (Affymetrix 6.0 [$N = 8640$], Perlegen-Affymetrix [$N = 1238$], Illumina Human Quad Bead 660 [$N = 1439$], Affymetrix Axiom [$N = 3144$], Illumnia GSA [$N = 5938$] and Illumina Omni Express 1 M [$N = 238$]), as well as sequence data from the Netherlands reference genome project GONL (BGI full sequence at $12 \times$ ($N = 364$)[48]. For each genotyping array samples were removed if they had a genotype call rate above 90%, gender-mismatch occurred or if heterozygosity (Plink F statistic) fell outside the range of $-0.10$ to $0.10$. SNPs were removed if they were palindromic AT/GC SNPs with a MAF range between 0.4 and 0.5, if the MAF was below 0.01, if Hardy Weinberg Equilibrium (HWE) had $p < 10^{-5}$, and if the number of Mendelian errors was greater than 20 and the genotype call rate was <0.95. After QC the six genotyping arrays were aligned to the GONL reference set (V4) and SNPs were removed if the alleles mismatched with this reference panel or the allele frequency different more than 0.10 between the genotyping array and this reference set.

The data from the six genotyping chips were subsequently merged into a single dataset (1,781,526 SNPs). Identity-by-decent (IBD) was estimated with PLINK[49] and KING[50] for all individual pairs based on the ~10.6 K SNPs in common across the arrays. Next IBD was compared to expected family relations and individuals were removed in the event of a mismatch. Prior to imputation to the GONL reference data[51,52] the duplicate monozygotic pairs ($N = 3032$) or trios ($N = 7$) and NTR GONL samples ($N = 364$) were removed and the data was cross-array phased using MACH-ADMIX[53]. Post-imputation the NTR GONL samples and the duplicated MZ pairs and trios were returned to the dataset. Filtering of the imputed dataset included the removal of SNPs that were significantly associated with a single genotyping chip ($p < 10^{-5}$), had HWE $p < 10^{-5}$, the Mendelian error rate > mean + 3 SD, or imputation quality ($R^2$) below 0.90. The final cross-platform imputed dataset included 1,314,639 SNPs, including 20,792 SNPs on the X-chromosome.

The cross-platform imputed data was aligned with PERL based HRC or 1000G Imputation preparation and checking tool (version 4.2.5; https://www.well.ox.ac.uk/~wrayner/tools). The remaining 1,302,481 SNPs were phased with EAGLE[54] for the autosomes, and SHAPEIT[55] for chromosome X and then imputed to 1000 Genomes Phase 3 (1000GP3 version 5)[56] on the Michigan Imputation server using Minimac3 following the standard imputation procedures of the server[57]. PC analysis (PCA) was used to project the first 10 PCs of the 1000 genomes references set population on the NTR cross-platform imputed data using SMARTPCA[58]. Ancestry outliers (non-Dutch ancestry; $N = 1823$) were defined as individuals with PC values outside the European/British population range[59]. After ancestry outlier removal the first 10 PCs were recalculated.

**Curation of metabolite loci.** In October 2018 PubMed and Google Scholar were searched to identify published GWA and (exome-) sequencing studies on metabolomics or fatty acid metabolism in blood samples using ${}^1$H-NMR, mass spectrometry or gas chromatography-based methods. In the period of November 2008 to October 2018 40 GWA or (exome-) sequencing studies on blood metabolomics in European samples were published (Supplementary Table 1). The genome-wide significant ($p < 5 \times 10^{-8}$) metabolite-SNP associations of all studies were extracted, including only those observations for autosomal SNPs and reporting SNP effect sizes and $p$ values based on the summary statistics excluding NTR samples[46,47]. In the 40 studies, 242,580 metabolite-SNP or metabolite ratio-SNP associations were reported. These associations included 1804 unique metabolites or ratios and 49,231 unique SNPs (Supplementary Data 1). For all metabolites their Human Metabolome Database (HMDB)[2], PubChem[60], Chemical Entities of Biological Interest[61] and International Chemical Identifier[62] identifiers were retrieved. Information with regards to the super class, class and subclass of metabolites was extracted from HMDB. If no HMDB identifier was available and categorization information could not be extracted, super class, class and subclass were provided based on expert opinion. Excluding the ratios and unidentified metabolites, 953 metabolites were classified into 12 super classes, 43 classes or 77 subclasses (Supplementary Data 1). Based on the metabolite identifiers we also extracted the log(S) value for each metabolite to assess the hydrophobicity of the metabolites. The log(S) value represents the log of the partition coefficient between 1-octanol and water, two fluids that hardly mix. The partition coefficient is the ratio of concentrations in water and in octanol when a substance is added to an octanol-water mixture and hence indicates the hydrophobicity of a compound. Thus, we classified a metabolite as hydrophobic if it is more hydrophobic than 1-octanol, and as hydrophilic otherwise (Supplementary Data 1).

The rsIDs or chromosome-base pair positions of the 49,231 unique SNPs were reported by different genome builds or dbSNP maps[63], therefore we lifted all SNPs to HG19 build 37[64], after which 43,830 unique SNPs remained (Supplementary Fig. 1; Supplementary Data 1). All biallelic metabolite SNPs were extracted from our 1000GP3 data, which excluded 295 triallelic SNPs, and 4256 SNPs that could not be retrieved from 1000GP3. Next, MAF > 1% (2067 SNPs removed), $R^2 > 0.70$ (2002 SNPs) and HWE $p < 10^{-4}$ (72 SNPs) filtering was performed, resulting in 35,138 metabolite SNPs for NTR participants (Supplementary Fig. 1). Next, we

created two super class-specific lists of metabolite loci and two not-superclass lists of metabolite loci. To create a list of loci associated with the 652 unique metabolites classified as lipids and lipid-like molecules (e.g., lipids), we clumped (PLINK version 1.9) all 112,760 lipid-SNP associations using an LD-threshold ($r^2$) of 0.10 in a 500 kb radius in 2500 unrelated individuals (Supplementary Fig. 1). Clumping identified 482 lead SNPs, or loci for lipids. An additional 12,169 SNPs were identified as LD-proxies for the lipid-loci (Supplementary Fig. 1). To obtain the not-superclass list of lipid loci the 12,651 lipid loci and proxies were removed from the list of all metabolite-SNP associations and the resulting list was clumped to obtain the 598 non-superclass loci (Supplementary Fig. 1). The same clumping procedure was applied to the 26,352 organic acid-SNP associations, identifying 398 organic acids loci, 10,781 organic acid LD-proxies, and 687 non-superclass loci (Supplementary Fig. 1).

**Construction of genetic relationship matrices.** In total six weighted GRMs were constructed, which were corrected for uneven and long-range LD between the SNPs (LDAK version 4.9[26,27]). In Supplementary Note 3, the use of weighted versus unweighted GRMs is compared using simulations. Two of the GRMs used the cross-platform imputed dataset as backbone and the other four GRMs were based on SNPs extracted from the 1000GP3 imputed data. Before calculating the first GRM, the autosomal SNPs of the cross-platform imputed dataset were filtered on MAF (<1%) and all lipid and organic acid loci, their LD-proxies and 50 kb surrounding both types of SNPs were removed (see curation of metabolite loci; Supplementary Fig. 1). The LDAK GRM was created after removal of the 50 kb surrounding the lipid and organic acid loci and their LD-proxies (as obtained by the clumping procedure as described above) and included 434,216 SNPs (Supplementary Fig. 1). The V(G1) variance component in the GREML analyses is based on this GRM (see heritability analyses; Fig. 1). The V(G2) variance component in the GREML analyses is based on the LDAK GRM including all autosomal SNPs with a MAF greater than 1% included on the cross-platform imputed dataset (447,794 SNPs), where ancestry outliers were removed, and genome sharing was set to zero for all individual pairs sharing less than 0.05 of their genome[19] (Fig. 1). Depending on the metabolite the V(G3) variance component in the GREML analyses is either based on an LDAK GRM of the 1000GP3 extracted lipid loci (479 SNPs) or the organic acid loci (397 SNPs), as obtained after the clumping procedure as described above (Supplementary Fig. 1; Fig. 1). Finally, depending on the metabolite either the not-lipid LDAK GRM (596 SNPs) or the not-organic acid LDAK GRM (683 SNPs) provided the V(G4) variance component in the GREML analyses (Supplementary Fig. 1; Fig. 1). The not-class metabolite loci on which the LDAK GRMs were build were obtained by the clumping procedure as described above (Supplementary Fig. 1). Supplementary Data 1 indicates for each listed SNP if it was included in any of the class-specific or not-class LDAK GRMs.

**Heritability analyses.** Mixed linear models[19], implemented in the GCTA software package (version 1.91.7)[18], were applied to compare three models including a variable number of covariates. Supplementary Table 5 gives the three different models, full descriptions of the covariates and model comparison have been given in Supplementary Note 4. The most parsimonious model was chosen for further analyses (full results in Supplementary Table 6). This final model included the first ten genetic PCs for the Dutch population, genotyping chip, sex, and age at blood draw as covariates. For metabolites of the Nightingale Health ${}^1$H-NMR and Biocrates platform, measurement batch was included as covariate.

The final four-variance component model, including four GRMs, allows for the estimation of the proportion of variation explained by superclass-specific significant metabolite loci and non-superclass significant metabolite loci. The first two-variance components in the four-variance component model (Fig. 1), V(G1) and V(G2) allow for the estimation of the additive genetic variance effects captured by genome-wide SNPs ($h^2_g$) and the additive genetic effects associated with pedigree ($h^2_{ped}$)[19,65], and V(G3) and V(G4) capture the additive genetic effect associated with class-specific ($h^2_{Class-hits}$) and not-class ($h^2_{Notclass-hits}$) metabolite loci. Based on the four-variance component model, three additional heritability estimates can be calculated: the total variance explained by significant metabolite loci ($h^2_{Metabolite-hits}$) consists of the sum of $\frac{V(G3)}{Vp}$ and $\frac{V(G4)}{Vp}$, where Vp is the phenotypic variance, $h^2_{SNP}$ is defined as the sum of $\frac{V(G1)}{Vp}$, $\frac{V(G3)}{Vp}$ and $\frac{V(G4)}{Vp}$, and the total variance explained ($h^2_{total}$) is defined as the sum of $\frac{V(G1)}{Vp}$, $\frac{V(G2)}{Vp}$, $\frac{V(G3)}{Vp}$, and $\frac{V(G4)}{Vp}$ (Fig. 1). We note that the total variance explained by genetic factors may also include influences of the shared environment, dominance and epistasis, which may result in upward bias of the $h^2_{total}$ estimates[19,20]. This bias is expected to arise by the presence of closely related participants, who may share these effects, in addition to the additive genetic effects. To calculate the standard errors (s.e.'s) for the composite variance estimates, we have randomly sampled 10,000 new variances from the parameter variance-covariance matrices of the V(G1), V(G3), and V(G4) GRMs for each metabolite. Random sampling was performed in R by creating 10,000 multivariate normal distributions (mvrnorm function in MASS package version 7.3-50[66]) based on the original means and variance/covariance matrices. The s.e.'s of the specific ratio of interest were then based on the standard deviation of the ratio of interest across 10,000 samples. The four-variance component models included variance components that were not constrained to be positive, thus

allowing for negative $h^2_{SNP}$ and $h^2_{Metabolite-hits}$ estimates. All four-variance component models applied the --reml-bendV flag where necessary to invert the variance-covariance matrix $V$ if $V$ was not positive definite, which may occur when variance components are negative[67]. Finally, we calculated the log likelihood of a reduced model with either V(G3), V(G4), or both dropped from the full model and calculated the LRT and $p$ value (Supplementary Data 3).

**Mixed-effect meta-regression analyses.** To investigate differences in heritability estimates among metabolites of different classes we applied mixed-effect meta-regression models as implemented in the metafor package (version 2.0-0) in R (version 3.5.1)[68]. Here, we tested for the moderation of heritability estimates by metabolite class and metabolomics platform on all 361 successfully analyzed metabolites. We included a matrix combining the phenotypic correlations (Supplementary Data 10) and the sample overlap (Supplementary Data 9) between the metabolites as random factor to correct for dependence among the metabolites and participants. This matrix includes the sample size of the metabolite on the diagonal, with the off-diagonal computed by $\frac{N_{1,2}}{\sqrt{n_1 * n_2}} * r$ (Supplementary Data 11), where $N_{1,2}$ is the sample overlap between the metabolites, $n_1$ is the sample size of metabolite one, $n_2$ is the sample size of metabolite two and $r$ is the phenotypic (Spearman's rho) correlation between the metabolites. In all mixed-effect meta-regression analyses we obtained the robust estimates based on a sandwich-type estimator, clustered by the metabolites included in the models to correct for the sample overlap among the different metabolites[69]. First, we used multivariate mixed-effect meta-regression models to simultaneously estimate the effect of metabolite class and metabolomics platform on the $h^2_{total}$, $h^2_{SNP}$, and the $h^2_{Metabolite-hits}$, as well as the $h^2_{Class-hits}$ and $h^2_{Notclass-hits}$ estimates. Subsequently, to separately assess the effect of the number of carbon atoms or double bonds in the fatty acyls chains of phosphatidylcholines and TGs univariate models were fitted, as follow-up. To account for multiple testing the p-values were adjusted with the with the FDR[70] using the p.adjust function in R. Multiple testing correction was done separately for the univariate and the multivariate models.

**Reporting summary.** Further information on research design is available in the Nature Research Reporting Summary linked to this article.

## Data availability

The curated list of all published metabolite-SNP associations is included in Supplementary Data 1 and is publicly available through the BBMRI—omics atlas (http://bbmri.researchlumc.nl/atlas/#data). All information on the metabolites in this study are in Supplementary Data 2; with full summary statistics for the four-variance component models included in Supplementary Data 3. The Nightingale Health metabolomics data may be requested through BBMRI-NL (https://www.bbmri.nl/Omics-metabolomics). All (other) data may be accessed, upon approval of the data access committee, through the NTR (ntr.fgb@vu.nl).

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

## Acknowledgements

We thank all twins and family members for their participation. We thank P.M. Visscher (University of Queensland) for his helpful comments and C.V. Dolan (Vrije Universiteit Amsterdam) for critically reading and commenting on the final version of the paper. Preliminary analyses of this paper were included in a presentation at the 46th Annual Meeting of the Behavioral Genetics Association (BGA) in June 2016 (abstract in Behav. Genet. (2016) 46:785–786), and a presentation at the 49th Annual Meeting of the BGA in June 2019 (abstract forthcoming). This work was performed within the framework of the BBMRI Metabolomics Consortium funded by BBMRI-NL, a research infrastructure financed by the Dutch government (NWO, nos. 184.021.007 and 184.033.111). The European Network of Genomic and Genetic Epidemiology (ENGAGE) contributed to funding to perform the Biocrates Absolute-IDQ™ p150 metabolomics measurements (European Union Seventh Framework Program: FP7/2007–2013, grant number 201413). Analyses were supported by the Netherlands Organization for Scientific Research: Netherlands Twin Registry Repository: researching the interplay between genome and environment (480-15-001/674); the European Union Seventh Framework Program (FP7/2007–2013): ACTION Consortium (Aggression in Children: Unraveling gene–environment interplay to inform Treatment and InterventiON strategies; Grant number 602768). Genotyping was made possible by grants from NWO/SPI 56-464-14192, Genetic Association Information Network (GAIN) of the Foundation for the National Institutes of Health, Rutgers University Cell and DNA Repository (NIMH U24 MH068457-06), the Avera Institute, Sioux Falls (USA) and the National Institutes of Health (NIH R01 HD042157-01A1, MH081802, Grand Opportunity grants 1RC2 MH089951 and 1RC2 MH089995) and European Research Council (ERC-230374). EMIF-AD has received support from the EU/EFPIA Innovative Medicines Initiative Joint Undertaking EMIF grant agreement no. 115372. DIB acknowledges her KNAW Academy Professor Award (PAH/6635). M. Bartels is supported by an ERC consolidator grant (WELL-BEING 771057 PI Bartels). Jv.D. is supported by the NWO-funded X-omics project (184.034.019).

## Author contributions

Nightingale Health metabolomics data: H.E.S., M. Beekman, P.E.S. and C.Mv.D. Leiden ¹H-NMR metabolomics data: K.Wv.D. and A.V. UPLC-MS lipidomics data: A.C.H. and T.H. EMIF-AD data: Ad.B. and P.J.V. Genotype data: J.J.H., A.A., and I.O.F. NTR Biobank data: G.W. and E.J.Cd.G. Metabolomics preprocessing: R.P., H.H.M.D. and F.A.H. Statistical analyses: F.A.H. and M.G.N. Wrote the paper: F.A.H., Jv.D., M. Bartels, M.G.N. and D.I.B. All authors critically read and commented on the paper.

## Competing interests

The authors declare no competing interests.

## Additional information

## BBMRI Metabolomics Consortium

J.J.H. Barkey Wolf[10], D. Cats[10], N. Amin[11], J.W. Beulens[13,14], J.A. van der Bom[15,16,17,18], N. Bomer[19], A. Demirkan[11], J.A. van Hilten[20], J.M.T.A. Meessen[21], M.H. Moed[10], J. Fu[22,23], G.L.J. Onderwater[24], F. Rutters[13], C. So-Osman[20], W.M. van der Flier[3,13], A.A.W.A. van der Heijden[25], A. van der Spek[11], F.W. Asselbergs[26], E. Boersma[27], P.M. Elders[2,28], J.M. Geleijnse[29], M.A. Ikram[11,30,31], M. Kloppenburg[18,32], I. Meulenbelt[10], S.P. Mooijaart[33], R.G.H.H. Nelissen[34], M.G. Netea[35,36], B.W.J.H. Penninx[2,37], C.D.A. Stehouwer[38,39], C.E. Teunissen[40], G.M. Terwindt[24], L.M. 't Hart[2,10,13,41,42], A.M.J.M. van den Maagdenberg[43,7], P. van der Harst[18], I.C.C. van der Horst[43], C.J.H. van der Kallen[38,39], M.M.J. van Greevenbroek[38,39], W.E. van Spil[44], C. Wijmenga[22], A.H. Zwinderman[45], A. Zhernikova[22], J.W. Jukema[46], H. Mei[10,47], M. Slofstra[22], M. Swertz[22], E.B. van den Akker[10,48,49], J. Deelen[10,50] & M.J.T. Reinders[48,49]

[13]Department of Epidemiology and Biostatistics, Amsterdam University Medical Center, Amsterdam, The Netherlands. [14]Julius Center for Health Sciences and Primary Care, University Medical Center Utrecht, Utrecht, The Netherlands. [15]Centre for Clinical Transfusion Research, Sanquin Research, Leiden, The Netherlands. [16]Jon J van Rood Centre for Clinical Transfusion Research, Leiden University Medical Centre, Leiden, The Netherlands. [17]TIAS, Tilburg University, Tilburg, The Netherlands. [18]Department of Clinical Epidemiology, Leiden University Medical Centre, Leiden, The Netherlands. [19]Department of Cardiology, University Medical Center Groningen, University of Groningen, Groningen, The Netherlands. [20]Center for Clinical Transfusion Research, Sanquin Research, Leiden, The Netherlands. [21]Department of Orthopedics, Leiden University Medical Centre, Leiden, The Netherlands. [22]Department of Genetics, University Medical Center Groningen, University of Groningen, Groningen, The Netherlands. [23]Department of Pediatrics, University Medical Center Groningen, University of Groningen, Groningen, The Netherlands. [24]Department of Neurology, Leiden University Medical Center, Leiden, The Netherlands. [25]Department of General Practice, The EMGO Institute for Health and Care Research, VU University Medical Center, Amsterdam, The Netherlands. [26]Department of Cardiology, Division Heart and Lungs, University Medical Center Utrecht and the Julius Center for Health Sciences and Primary Care, University Medical Center Utrecht, Utrecht, The Netherlands. [27]Thorax Centre, Erasmus Medical Centre, Rotterdam, The Netherlands. [28]Department of General Practice and Elderly Care Medicine, VU University Medical Center, Amsterdam, The Netherlands. [29]Division of Human Nutrition and Health, Wageningen University, Wageningen, The Netherlands. [30]Department of Radiology, Erasmus University Medical Center Rotterdam, Rotterdam, The Netherlands. [31]Department of Neurology, Erasmus University Medical Center Rotterdam, Rotterdam, The Netherlands. [32]Department of Rheumatology, Leiden University Medical Center, Leiden, The Netherlands. [33]Department of Internal Medicine, Division of Gerontology and Geriatrics, Leiden University Medical Centre, Leiden, The Netherlands. [34]Department of Orthopaedics, Leiden University Medical Center, Leiden, The Netherlands. [35]Department of Internal Medicine, Radboud Center for Infectious Diseases, Radboud University Medical Center, Nijmegen, Netherlands. [36]Department for Genomics & Immunoregulation, Life and Medical Sciences Institute (LIMES), University of Bonn, Bonn, Germany. [37]Department of Psychiatry, VU University Medical Center, Amsterdam, The Netherlands. [38]Department of Internal Medicine, Maastricht University Medical Center (MUMC+), Maastricht, The Netherlands. [39]School for Cardiovascular Diseases (CARIM), Maastricht University, Maastricht, The Netherlands. [40]Neurochemistry Laboratory, Clinical Chemistry Department, Amsterdam University Medical Center, Amsterdam Neuroscience, Amsterdam, The Netherlands. [41]Department of Cell and Chemical Biology, Leiden University Medical Center, Leiden, The Netherlands. [42]Department of General practice, Amsterdam University Medical Center, Amsterdam, The Netherlands. [43]Department of Critical Care, University Medical Center Groningen, Groningen, The Netherlands. [44]UMC Utrecht, Department of Rheumatology & Clinical Immunology, Utrecht, The Netherlands. [45]Department of Clinical Epidemiology, Biostatistics, and Bioinformatics, Academic Medical Centre, University of Amsterdam, Amsterdam, The Netherlands. [46]Department of Cardiology, Leiden University Medical Center, Leiden, The Netherlands. [47]Sequencing Analysis Support Core, Leiden University Medical Center, Leiden, The Netherlands. [48]Leiden Computational Biology Center, Leiden University Medical Center, Leiden, The Netherlands. [49]Department of Pattern Recognition and Bioinformatics, Delft University of Technology, Delft, The Netherlands. [50]Max Planck Institute for Biology of Ageing, Cologne, Germany

