## [Peer Review File · Nature Communications]

Reviewers' Comments:

Reviewer #1:

Remarks to the Author:

Hagenbeek et al perform a heritability analysis of about 400 metabolites. First, they estimate averages for h^2 , h^2 SNP and h^2 GWAS, the total heritability, snp heritability and heritability explained by genome-wide significant SNPs. Then they apply a state of the art method, BayesS, to estimate for each metabolite, a selection parameter S.

I'm sorry to say, but in my view, this paper has a number of moderate to serious methodological problems.

Major comments:

1 - There are a number of problems with the heritability estimation.

a - It is very concerning that many (20%) of the analyses did not converge. By discarding these, you can be introducing biases - you are not estimating averages over all metabolites, but just those which converge. Ideally you would identify what factors cause failure, and then work out quality control filters (which could be applied to all metabolites) to exclude these. Also, I don't agree with your approach to prefer using constrained variances (ie not allow variances outside 0-1). This can cause boundary issues - if you force small variances to be positive, you will tend to over-estimate, and vice versa. So i think you should always use non-constrained.

b - I wonder if the very small SDs you observed are a consequence of the constraining? But in any case, I do not see it as an acceptable fudge to up-truncate SDs because they appear unrealistically small (instead, as with 1a, I think you have to get to the heart of the problem).

c - SNP heritabilities are ideally estimated from an unrelated dataset. The Zaitlen method you use is designed when there are complex patterns of relatedness, such that pruning will drastically reduce sample size. But here it seems you have fairly simple relationships - many of the closely related are twins? So it would seem better to first divide your data into two subgroups, such that individuals in each subgroup are unrelated (were all your individuals pairs of twins, you could simply put the first of each pair in group a, the second in group b - as that is not the case, you could instead prune individuals with say a cutoff of 0.05 to make group a, then prune the individuals who were discarded to make group b). It seems you should be able to construct two reasonably sized groups, so that you can then analyze each separately (without requiring Zaitlens truncated kinship method), and meta-analyse the two sets of estimates.

d - For a particular phenotype, h^2 gwas is generally defined as the variance explained by variants detected through gwas (ie SNPs achieving gwide significance) for THAT phenotype. So to estimate h^2 gwas, you must first perform a GWAS (or identify a previously performed GWAS for that metabolite), identify the significant SNPs, then compute how much variance they explained. Hagenbeek instead define h^2 GWAS as the variance explained by a constant set of 500 snps - these SNPs have been picked as they have been determined to be gwide significant for A metabolite (i.e., of all the metabolite studies consider, each SNP is associated for at least one), but for a given metabolite the 500 SNPs will almost certainly contain only some of the QTLS, and contain many SNPs which are not QTLS. Therefore, the h^2 GWAS estimates of Hagenbeek will likely be substantial underestimates for some metabolites (those for which there is a strongly associated snp which is not one of the 500 considered) or over-estimates for others (where the estimate of h^2 gwas is capturing the variance of lots of non-significant SNPs).

e - Related to 1d, I do not see what is preventing you from estimating h^2 gwas the standard way - ie performing a GWAS for each metabolite. This would have the advantage that then when estimating h^2 SNP, you could include top SNPs as covariates (as shown in your Ref81, when you

have large effect loci, these violate the fundamental assumption of an infinitesimal model, so it is preferable to include them as fixed effects)

f - Even if the 500 SNPs were the significant SNPs for each metabolite (or included them), I'm not sure the mixed model approach will be a good way to estimate their total contribution. The usual way to estimate their variance would be either through classical least squares (perhaps adjusting for possible overfitting), or mixed-model with a single GRM (which if restricting to gwide significant, should automatically adjust). Hagenbeek instead using 2 or 3 component mixed models. But in this, the gwide GRM will contain many SNPs in high LD with the 500 SNPs, so the gwide GRM will "suck" up some of the variance explained by the 500 - at a guess, the resulting estimates of h^2_{gwas} would be at least 50% lower than their true value.

g - As you are averaging over many metabolites, you should consider their correlations. E.g., it could be that 50 metabolites are highly correlated, in which case this group would have (arguably) an undue effect on the averages, and would certainly affect the precision of the mean. The bimodal nature of Supplementary Figure 8a indicates there are strong correlations. Therefore, would be good to see a consideration of a approx uncorrelated / less correlated subset of metabolites.

h - A common theme of the paper is a lack of SDs which make it hard to tell if differences are significant.

- BayesS is a state-of-the-art software, but the authors use of it is unusual. Zeng (the BayesS author) applied it to gwide data (ie 100 thousands of SNPs) in order to identify the gwide relationship between effect sizes and MAF; Hagenbeek apply it to only 500 SNPs. If these 500 SNPs were picked at random, it might be ok (but is it really ok to extrapolate about gwide relationships from just 500 snps?). But instead these SNPs are selected as potential QTLs. This is very likely to bias the analysis (e.g., if they were QTLs, then this places an artificial lower bound on the magnitude of their effect sizes). Further, as per 1h and 1g, without SDs, it is not obvious whether the difference in estimates of selection between classes is significant (e.g, the ranges clearly overlap), and I am concerned that the correlations between metabolites are affecting results.

Minor comments:

- The abstract does not contain SDs (or conf ints) or total sample size; the main text does not contain SDs, and when you mention the sample sizes, could you mention a total please (and not just the range of the sample sizes across the four platforms). Also, please put number of snps in main text.

- You state that there are significant differences across classes, but I did not see where you tested for significance (really sorry if I missed it).

- Please can you provide a few representative histograms of what metabolite levels look like. For your analyses, ideally they would be approximately normal (or maybe bimodal, as then the relative values would still be accurate) - so could you make a few histograms for a few metabolites picked at random to show whether they are. If some are not, you should consider excluding the least-normal looking ones (or outliers individuals).

- I would define the hidden heritability as $1-h^2_{GWAS}/h^2$ (as in what proportion of h^2 is hidden across small effect variants) rather than your definition as $1-h^2_{GWAS}/h^2_{SNP}$. Also, I think in L82 you must include the denominator (ie missing heritability is $(h^2-h_{SNP})/h_{SNP}$).

- L383 - "heterozygosity fell outside the range of -0.075 – 0.075" - heterozygosity is greater than 0, so are these values relative to mean?

- L404 - what threshold for unrelated individuals were set to zero. Did you use 5% (?) like zaitlen?

- Supplementary Figure 6 - very good to check consistency between bayess and gcta est of h2gwas, but these results seems to contradict your claim that BayesS is not affected by relatedness (the figure shows a systematic upwards bias - although it could be that estimates from BayesS are correct, and the mixed model estimates are too small, as per 1d)

- Supp note 1 - it's very good to test bayess through simulation - but it seems the heritabilities you used in your simulations (10, 20, 40, 60, 80%) are much higher than those observed for real data.

Very minor comments:

- In methods, you explain you use LDAK weighted kinships (which obviously I support). But in the main text, you only mention GCTA (which you use for the REML estimation). When you say in L104 that you use GCTA, I think most people will assume you use the GCTA Method (ie GCTA Kinships), rather than just the GCTA REML solver. Therefore, consider changing this (perhaps say used "GCTA and LDAK", or "GCTA with LDAK Kinships", or something similar). Or, considering you only use GCTA for its REML solver, you could instead use the REML solver in LDAK.

- In the abstract, I think it would be better to state the absolute values, then relative values - atm, you state abs h2total, then relative h2gwas & relative h2snp, then abs h2snp & h2gwas

Signed Doug Speed

Reviewer #2:

Remarks to the Author:

In their work "Comprehensive consideration of the genetic architecture of the human metabolome" Hagenbeek et al presents a characterisation of heritability estimates across 357 metabolic features from human blood. The authors have indeed done a lot of work to put it together. I find the work important and I think it will serve as a basis for many scientists working on genetics of human metabolism.

I have few comments for revision:

1. The authors should change from using "metabolites" to something that describes all 357 measures better, such as "metabolic measures". For example, if one considers the Nightingale platform, many of the measures are not exactly metabolites (for example "Ratio ApoA1/ApoB").
2. The abstract requires further polishing. It is rather difficult to follow.
3. Symbols for indicating different heritability's are usually in italic, but often not. Please be concise.
4. I find the last part of the paper that deals with selection problematic. In most cases the causal variant or haplotype is not known. Therefore the conclusions drawn from the allele frequency and effect estimate is based on unknown linkage disequilibrium between the associated variant and the

true causal allele and because that relationship is unknown, any conclusions drawn from the MAF and beta can be, and most likely are, erroneous. Unless the authors can convincingly justify these analyses, they must be removed from the paper.

5. How is the GW-loci GRM affected by LD between the SNPs that were used to build it? Many of the SNPs are from the same genetic locus (Figure 1) and can describe the same association signal but just have another metabolic measure associated with it. If you do LD pruning on those 497 SNPs, how many are left in if you use r^2 cutoff of 0.1? Does it affect the results?

Point-by-point response to the reviewer comments:

The genomic architecture of blood metabolites based on a decade of genome-wide analyses

Fiona A. Hagenbeek^{1,2*}, René Pool^{1,2}, Jenny van Dongen^{1,2}, Harmen H.M. Draisma¹, Jouke Jan Hottenga¹,
Gonneke Willemsen¹, Abdel Abdellaoui¹, Iryna O. Fedko¹, Anouk den Braber^{1,3}, Pieter Jelle Visser^{3,4}, Eco
J.C.N. de Geus^{1,2,5}, Ko Willems van Dijk⁶, Aswin Verhoeven⁷, H. Eka Suchiman⁸, Marian Beekman⁸, P. Eline
Slagboom⁸, Cornelia M. van Duijn⁹, BBMRI-NL Consortium¹⁰, Amy C. Harms¹¹, Thomas Hankemeier¹¹,
Meike Bartels^{1,2,5}, Michel G. Nivard^{1,2,5*‡} and Dorret I. Boomsma^{1,2,5*‡}

Reviewer #1

Major comments:

1 - There are a number of problems with the heritability estimation.

a - It is very concerning that many (20%) of the analyses did not converge. By discarding these, you can be introducing biases - you are not estimating averages over all metabolites, but just those which converge. Ideally you would identify what factors cause failure, and then work out quality control filters (which could be applied to all metabolites) to exclude these. Also, I don't agree with your approach to prefer using constrained variances (i.e. not allow variances outside 0-1). This can cause boundary issues - if you force small variances to be positive, you will tend to over-estimate, and vice versa. So I think you should always use non-constrained.

Thank you for suggesting unconstrained variances to address the convergence issues we've been having and pointing out the biases these convergence issues could introduce to our methodological framework. The analyses in the revised version of our manuscript have been based on the unconstrained variances. Switching to unconstrained variances has increased our successful analysis rate to 97.8%. The remainder

of the non-successful analyses could be attributed to non-invertible variance-covariance matrices or non-
convergence of analyses after successful ‘bending’ to allow for inversion of the variance-covariance
matrix. Additionally, we appreciate your suggestion of including factors which might contribute to non-
convergence as quality control filters to be applied to all metabolites and will certainly keep this in mind
for potential future work. However, we felt adjusting the quality control filters based on observed results
in the same study would be ‘bad practice’.

*b - I wonder if the very small SDs you observed are a consequence of the constraining? But in any case, I do*
*not see it as an acceptable fudge to up-truncate SDs because they appear unrealistically small (instead, as*
*with 1a, I think you have to get to the heart of the problem).*

We appreciate your suggestion that the very small observed SE’s could be due to the use of constrained
variances. In this revised manuscript we have based all estimates on the unconstrained variance
component and have additionally adjusted how we’ve calculated the SE of the h^2_{SNP} , which no longer
results in the unrealistically small SE’s. Currently we use the following procedure to calculate the SE for
h^2_{SNP} : we randomly sampled 10,000 instances from the parameter variance-covariance matrices for each
metabolite. The s.e.’s of the specific ratio of interest were then based on the standard deviation of the
ratio of interest across 10,000 samples. The same approach has been used for the calculation of our
redefined $h^2_{GW-loci}$ SE’s.

*c - SNP heritabilities are ideally estimated from an unrelated dataset. The Zaitlen method you use is*
*designed when there are complex patterns of relatedness, such that pruning will drastically reduce sample*
*size. But here it seems you have fairly simple relationships - many of the closely related are twins? So it*

*would seem better to first divide your data into two subgroups, such that individuals in each subgroup are*
*unrelated (were all your individuals pairs of twins, you could simply put the first of each pair in group a, the*
*second in group b - as that is not the case, you could instead prune individuals with say a cutoff of 0.05 to*
*make group a, then prune the individuals who were discarded to make group b). It seems you should be*
*able to construct two reasonably sized groups, so that you can then analyze each separately (without*
*requiring Zaitlens truncated kinship method), and meta-analyze the two sets of estimates.*

Thank you for suggesting a possible alternative to the Zaitlen-method for including large numbers of twins
in the analyses. However, we fear that by splitting the data into two (or more) groups of unrelated
individuals and meta-analyzing them together, bias will be introduced at the meta-analytical level, as
these groups will not be independent from one another. And while many of the closely related individuals
in our dataset are indeed twins (63.4% of the complete sample), our data also include parents, siblings or
offspring of twins and families thus are sometimes more complex. Moreover, twin families within this data
might well share (distant) kinship. Overall, we feel that meta-analyzing two or more groups would not be
the best course of action and continued to apply the Zaitlen method in the revised manuscript.

*d - For a particular phenotype, h^2_{gwas} is generally defined as the variance explained by variants detected*
*through gwas (ie SNPs achieving gwide significance) for THAT phenotype. So to estimate h^2_{gwas} , you must*
*first perform a GWAS (or identify a previously performed GWAS for that metabolite), identify the*
*significant SNPs, then compute how much variance they explained. Hagenbeek instead define h^2_{GWAS} as*
*the variance explained by a constant set of 500 snps - these SNPs have been picked as they have been*
*determined to be gwide significant for A metabolite (i.e., of all the metabolite studies consider, each SNP is*
*associated for at least one), but for a given metabolite the 500 SNPs will almost certainly contain only*

*some of the QTLs, and contain many SNPs which are not QTLs. Therefore, the h2GWAS estimates of*
*Hagenbeek will likely be substantial underestimates for some metabolites (those for which there is a*
*strongly associated snp which is not one of the 500 considered) or over-estimates for others (where the*
*estimate of h2gwas is capturing the variance of lots of non-significant SNPs).*

We highly appreciate the reviewer comment 1d and 1e where the reviewer indicates his suspicion that the
published estimates for $h^2_{GW-loci}$ will likely be under- or overestimated when including the 500 SNPs
reported to be associated with *any* metabolite. Instead reviewer 1 suggests in these comments to perform
GWAS analyses on each of the metabolites and use the subsequent significantly identified SNPs to
calculate the explained variances for each metabolite separately. Unfortunately, the sample size available
to us is quite small ($N = 5,117$) and varies greatly across platforms, ranging from 1,448 observations on the
Biocrates platform to 4,227 observations on the Nightingale Health platform. In addition, as the reviewer
also points out in comment 1g, many metabolites, that belong to the same biochemical class or pathway,
are highly correlated. In separately analyzing metabolites with different samples sizes, without accounting
for the correlation among the metabolites, we fear important genetic loci might be ‘missed’ for those
metabolites with a small sample size, which might be picked up by GWA on related metabolites with larger
sample sizes. Taking both these issues into consideration, we’ve decided against performing GWA analyses
on each of our metabolites.

While we have not performed GWA studies for each of our metabolites, we do really appreciate
the caution of reviewer 1 of biased $h^2_{GW-loci}$ estimates by the inclusion of SNPs not directly associated with
the target metabolite. We have attempted to address this issue in several steps. First, we have expanded
the list of 500 blood metabolite SNPs to include the blood metabolite-SNP associations in European
populations as published between November 2008 and October 2018 (40 studies in total), identifying
43,827 blood metabolite SNPs. Next, for all 942 unique metabolites (excl. ratios and unidentified

compounds) from the 40 studies, we retrieved the Human Metabolome Database (HMDB) identifiers.
 Based on the HMDB identifiers (or expert opinion if no HMDB identifier was retrieved) all metabolites
 were categorized (by the HMDB system) into 44 ‘classes’ (12 ‘super classes’). For each class represented in
 our metabolite data (12) we created GRMs capturing solely the metabolite loci for metabolites of this
 specific class (henceforth we will refer to the heritability estimates derived from these GRMs as $h^2_{GW-Class}$).
 Metabolite loci for each class were identified by including all published metabolite-SNP associations of the
 relevant class in a clumping procedure ($r^2 = 0.1$, radius = 500kb). The lead SNPs identified by the clumping
 procedure were then used as input to create LD-corrected GRMs in LDAK. To address whether the
 inclusion of associations for metabolites of a different biochemical class lead to biasing of the $h^2_{GW-loci}$
 estimates as included in the submitted manuscript, we used the same technique to create ‘non-class’
 GRMs, which used included the lead SNPs of all metabolite-SNP associations, excluding the relevant class
 loci and LD proxies (henceforth we will refer to the heritability estimates derived from these GRMs as
 $h^2_{GW-Notlass}$). This resulted in 4-variance component models, as resolved in the GCTA software.

Unfortunately, the proposed 4-variance component models for the 12 different classes of
 metabolites in our data had high degrees of non-convergence (37.9% total). Non-convergence of the 4-
 variance component models appears to be associated with the number of SNPs in the ‘class-specific’
 GRMs, with small GRMs displaying more convergence issues (**Table A**). Suggesting that metabolite-specific
 ‘hit GRMs’ would have resulted in similarly high non-convergence rates.

**Table A.** Convergence 4-variance component models for 12 metabolite classes

Metabolite class	Number of SNF LDAK GRM	Number of Metabolite:	Number conver analyses	Number of fai analyses
KetoAcidsAndDerivatives	20	2	0	2
OrganonitrogenCompounds	30	1	0	1
Glycerolipids	32	47	5	42
Sphingolipids	35	43	7	36

HydroxyAcidsAndDerivatives	49	5	0	5
OrganooxygenCompounds	51	9	1	8
Protein	80	3	0	3
Glycerophospholipids	85	145	104	41
SteroidsAndSteroidDerivatives	108	7	3	4
Lipoprotein	239	63	62	1
FattyAcyls	260	11	11	0
CarboxylicAcidsAndDerivatives	400	44	43	1

To address the convergence issues we've decided to recode the HMDB metabolite classes into
 HMDB metabolite super classes, reducing the 12 classes to 5 classes. Unfortunately, 3 of the 5 super
 classes only comprise of a single class and the number of SNPs in these GRMs will not increase. Since these
 analyses had previously shown poor convergence they were not been repeated. Combining classes into 2
 super classes with a larger number of SNPs in the GRM results in a convergence rate of 97.8% (**Table B**).

**Table B.** Convergence 4-variance component models for 5 metabolite superclasses

Metabolite Superclass	Metabolite classes included in superclass	Number of SNPs in LDAK GRM	Number of Metabolites	Number of convergent analyses	Number of failed analyses
Lipids and lipid-like molecules	FattyAcyls Glycerolipids Glycerophospholipids lipoprotein Sphingolipids Steroids and steroid derivatives	479	316	309	7
Organic acids and derivatives	Carboxylic acids and derivatives Hydroxy acids and derivatives Keto acids and derivatives	397	53	52	1
Organic nitrogen compounds	Organonitrogen Compounds	NA	1	NA	NA
Organic oxygen compounds	Organooxygen Compounds	NA	9	NA	NA
Protein	Protein	NA	3	NA	NA

*e - Related to 1d, I do not see what is preventing you from estimating h^2_{GW} the standard way - ie*
*performing a GWAS for each metabolite. This would have the advantage that then when estimating*
*h^2_{SNP} , you could include top SNPs as covariates (as shown in your Ref81, when you have large effect loci,*
*these violate the fundamental assumption of an infinitesimal model, so it is preferable to include them as*
*fixed effects)*

We highly appreciate the reviewer comment 1d and 1e where the reviewer indicates his suspicion that the
published estimates for $h^2_{GW-loci}$ will likely be under- or overestimated when including the 500 SNPs
reported to be associated with *any* metabolite. Instead reviewer 1 suggests in these comments to perform
GWAS analyses on each of the metabolites and use the subsequent significantly identified SNPs to
calculate the explained variances for each metabolite separately. Unfortunately, the sample size available
to us is quite small ($N = 5,117$) and varies greatly across platforms, ranging from 1,448 observations on
the Biocrates platform to 4,227 observations on the Nightingale Health platform. In addition, as the
reviewer also points out in comment 1g, many metabolites, that belong to the same biochemical class or
pathway, are highly correlated. In separately analyzing metabolites with different sample sizes, without
accounting for the correlation among the metabolites, we fear important genetic loci might be 'missed' for
those metabolites with a small sample size, which might be picked up by GWA on related metabolites with
larger sample sizes. Taking both these issues into consideration, we've decided against performing GWA
analyses on each of our metabolites.

*f - Even if the 500 SNPs were the significant SNPs for each metabolite (or included them), I'm not sure the*
*mixed model approach will be a good way to estimate their total contribution. The usual way to estimate*
*their variance would be either through classical least squares (perhaps adjusting for possible overfitting),*

*or mixed-model with a single GRM (which if restricting to gwide significant, should automatically adjust).*
*Hagenbeek instead using 2 or 3 component mixed models. But in this, the gwide GRM will contain many*
*SNPs in high LD with the 500 SNPs, so the gwide GRM will "suck" up some of the variance explained by the*
*500 - at a guess, the resulting estimates of h^2_{gwas} would be at least 50% lower than their true value.*

We concur with reviewer 1 that we had not addressed the LD among the multiple GRMs included in the 3-
variance component models and appreciate reviewer 1's insight in how this might impact the variance
explained by the GRM including the metabolite SNPs. As explained in the response to comment 1d, we
have now expanded the 3-variance component model to 4-variance component models. We have
accounted for the LD between the class-specific and non-class GRMs by removing the class-specific lead
SNPs and LD-proxies from the non-class GRM. To further address the LD among the class and non-class
GRMs with the GRM used to estimate h^2_{SNP} , we removed the lead SNPs, their LD-proxies and 50kb
surrounding each SNP from both GRMs from the GRM used to estimate the h^2_{SNP} .

*g - As you are averaging over many metabolites, you should consider their correlations. E.g., it could be*
*that 50 metabolites are highly correlated, in which case this group would have (arguably) an undue effect*
*on the averages, and would certainly affect the precision of the mean. The bimodal nature of*
*Supplementary Figure 8a indicates there are strong correlations. Therefore, would be good to see a*
*consideration of a approx uncorrelated / less correlated subset of metabolites.*

We greatly appreciate the reviewers insight with regards to the effect of the correlation structure among
metabolites on our analyses. As can be seen in **Supplementary Table 12** of the **Supplementary Tables** a
large number of the metabolites included in our study are indeed highly correlated. As the reviewer
mentions the calculate group means as reported in the main text and in **Tables 2-3** could be inflated due

to the underlying correlation structure of the metabolites averaged over (though of course their
underlying correlation structure is also why we average over these specific groups of metabolites). To
account of this we also applied multivariate mixed-model meta-regression models, corrected for the
sample overlap and correlation structure among the metabolites, to assess differences in metabolite
heritability.

*h - A common theme of the paper is a lack of SDs which make it hard to tell if differences are significant.*

We thank to reviewer for pointing out the lack of SEs throughout the main text of the manuscript. In the
previously submitted version of the manuscript we had neglected to include mentions of the SEs in the
main manuscript, though all SEs had been included in relevant the Supplementary Tables. The revised
version of the manuscript now includes the mean SE where relevant and the SEs for all individuals
metabolites are again included in the relevant Supplementary Tables.

*2 - BayesS is a state-of-the-art software, but the authors use of it is unusual. Zeng (the BayesS author)*
*applied it to gwide data (ie 100 thousands of SNPs) in order to identify the gwide relationship between*
*effect sizes and MAF; Hagenbeek apply it to only 500 SNPs. If these 500 SNPs were picked at random, it*
*might be ok (but is it really ok to extrapolate about gwide relationships from just 500 snps?). But instead*
*these SNPs are selected as potential QTLs. This is very likely to bias the analysis (e.g., if they were QTLs,*
*then this places an artificial lower bound on the magnitude of their effect sizes). Further, as per 1h and 1g,*
*without SDs, it is not obvious whether the difference in estimates of selection between classes is significant*
*(e.g, the ranges clearly overlap), and I am concerned that the correlations between metabolites are*
*affecting results.*

We would like to thank the reviewer with his honest assessment of our use of the BayesS method. We
concur that our choice to only include SNPs previously associated with metabolites, as opposed to
genome-wide SNPs, likely leads to biases in our analyses. In addition, we've noticed that the correlation
among the submitted BayesS results and the results as obtained with a larger dataset (based on all
published metabolite-SNP associations since 2008, as described in response to comment 1d) are lower
than anticipated ($r = 0.59$; $p = 1.315e-09$). This seems to suggest instability of the BayesS estimates across
relatively comparable runs, though at this time we cannot properly address or explain this. Therefore, in
accord with the suggestion of reviewer 2, we have removed the BayesS analyses from the manuscript.

Minor comments:

*3 - The abstract does not contain SDs (or conf ints) or total sample size; the main text does not contain SDs,*
*and when you mention the sample sizes, could you mention a total please (and not just the range of the*
*sample sizes across the four platforms). Also, please put number of snps in main text.*

We would like to thank the reviewer for pointing out these obvious caveats to the abstract and main text.
The revised main text now includes the SE where relevant, the SE has not been added to the abstract as
the abstract has been rewritten in such a manner that this would not be interpretable. The total sample
size has been included in the abstract, in the final paragraph of the introduction, in the first paragraph of
the 'characterization of the heritable influences on lipid and organic acid levels' subsection of the results,
in the first paragraph of the discussion, in the 'participants' subsection of the methods' and in **Table 1**. The
number of SNPs has been reported in the 'characterization of the heritable influences on lipid and organic
acid levels' subsection of the results and in the headings of **Figure 2** and **Figure 3**.

4 - You state that there are significant differences across classes, but I did not see where you tested for
significance (really sorry if I missed it).

We highly appreciate that the reviewer indicates it was unclear what the 'significant differences across
classes' was founded on. AS touched upon in responding to comment 1g, we have used multivariate
mixed-model meta-regression models to test for significant mean differences among metabolite classes
and lipid species. In the submitted version this was included in the final paragraph of the results section. In
an effort to clarify the use of these models we have now includes the results for each comparison after the
relevant descriptions of the mean heritabilities. We hope this improves the findability and readability of
the results section.

5 - Please can you provide a few representative histograms of what metabolite levels look like. For your
analyses, ideally they would be approximately normal (or maybe bimodal, as then the relative values
would still be accurate) - so could you make a few histograms for a few metabolites picked at random to
show whether they are. If some are not, you should consider excluding the least-normal looking ones (or
outliers individuals).

We thank the reviewer for his concerns with regards to the possible breach of the assumptions with
regards to normal distributions. We would like to point out that outlier detection of individuals has been
performed for all metabolites, as described in the methods. Additionally, all metabolites had been
transformed prior to analyses. Conform with previous studies using the same metabolomics platforms,
both ¹H-NMR platforms had been transformed using inverse rank normalization and were per definition
normally distributed, however, both MS platforms had been transformed using their natural logarithm.
while ln-normalization generally improves the skewedness of the distribution, some highly skewed

metabolites will still not have a perfectly normal distribution. Please find the histograms of 8 random
metabolites (4 from the Biocrates 'Bioc' platform and 4 from the Lipidomics 'Lip' platform) included here:

*6 - I would define the hidden heritability as $1-h^2_{GWAS}/h^2$ (as in what proportion of h^2 is hidden across*
*small effect variants) rather than your definition as $1-h^2_{GWAS}/h^2_{SNP}$. Also, I think in L82 you must include*
*the denominator (ie missing heritability is $(h^2-h_{snp})/h_{snp}$.*

We would like to thank reviewer one for sharing this thoughts on how to define 'hidden h^2 ' and for
pointing out the missing denominator on L82. The definitions as used in the submitted version were those
as defined in Witte, J. S., Visscher, P. M. & Wray, N. R. The contribution of genetic variants to disease
depends on the ruler. Nat. Rev. Genet. 15, 765–776 (2014). However, due to the use of unconstrained
variances in the current version of the manuscript, many of the estimates, especially for h^2_{SNP} are now
negative. Therefore, the ratios as defined by Witte et al. (2014) cannot be calculated for some of our
metabolites or result in uninterpretable estimates. For these reasons we have removed all references to
these ratios from the current manuscript and the abstract.

*7 - L383 - "heterozygosity fell outside the range of -0.075 – 0.075" - heterozygosity is greater than 0, so are*
*these values relative to mean?*

We would like to thank the reviewer for pointing out the confusing statement on L383 with regards to the
heterozygosity values. We use the PLINK F-statistic (or inbreeding coefficient) to determine
heterozygosity. This statistic can actually produce negative values, which commonly reflects sampling
error, though high negative values could indicate sample contamination (see
<http://zzz.bwh.harvard.edu/plink/ibdibs.shtml#inbreeding>). We have adjusted the 'genotyping' section in
the revised manuscript to reflect this more clearly. Additional changes have been made to this section to
reflect the changed made to the cleaning procedure when updating the genotype data freeze used on our
samples (to increase the sample size in the current analyses).

8 - L404 - what threshold for unrelated individuals were set to zero. Did you use 5% (?) like zaitlen?

We thank the reviewer for noticing we had neglected to include the threshold, the reviewer is correct in
assuming we had applied the same threshold was in the Zaitlen paper. We have adapted the relevant
section accordingly, which now includes the following statement: “for all individual pairs sharing less than
0.05 of their genome their sharing was set to zero”

9 - Supplementary Figure 6 - very good to check consistency between bayess and gcta est of h^2_{gwas} , but
these results seems to contradict your claim that BayesS is not affected by relatedness (the figure shows a
systematic upwards bias - although it could be that estimates from BayesS are correct, and the mixed
model estimates are too small, as per 1d)

We would like to thank the reviewer for pointing out the potential systematic upwards bias of the BayesS
$h^2_{GW-loci}$ estimates as compared with the GCTA $h^2_{GW-loci}$ estimates. However, we’ve noticed that the
correlation among the submitted BayesS results and the results as obtained with a larger dataset (based
on all published metabolite-SNP associations since 2008, as described in response to comment 1d) are
lower than anticipated ($r = 0.59$; $p = 1.315e-09$). Additionally, the correlation among the BayesS and GCTA
$h^2_{GW-loci}$ estimates are also much lower than anticipated on our previous results. The $h^2_{GW-loci}$ estimates of
updated 3-variance component models correlated 0.59 with the submitted BayesS estimates or 0.53 with
the updated BayesS estimates. This seems to suggest instability of the BayesS estimates across relatively
comparable runs, though at this time we cannot properly address or explain this. Therefore, in accord with
the suggestion of reviewer 2, we have removed the BayesS analyses from the manuscript.

*10 - Supp note 1 - it's very good to test bayess through simulation - but it seems the heritabilities you used*
*in your simulations (10, 20, 40, 60, 80%) are much higher than those observed for real data.*

We thank the reviewer for his concern with regards to the BayesS simulations. However, the BayesS
simulations had been included in **Supplementary Note 2**. The simulated heritability estimates in
**Supplementary Note 1** are used to compare the use of a weighted (LDAK) versus unweighted (GCTA) GRM
in our analyses and reflect the estimates for h^2_{SNP} . While h^2_{SNP} estimates of 40-80% could also be
considered quite high, similar values have been observed in our primary analyses.

Very minor comments:

*11 - In methods, you explain you use LDAK weighted kinships (which obviously I support). But in the main*
*text, you only mention GCTA (which you use for the REML estimation). When you say in L104 that you use*
*GCTA, I think most people will assume you use the GCTA Method (ie GCTA Kinships), rather than just the*
*GCTA REML solver. Therefore, consider changing this (perhaps say used "GCTA and LDAK", or "GCTA with*
*LDAK Kinships", or something similar). Or, considering you only use GCTA for its REML solver, you could*
*instead use the REML solver in LDAK.*

We thank the reviewer for pointing out that the distinction in the submitted manuscript with regards to
using GCTA only for its REML solver, while creating GRMs in LDAK, was unclear. As per the reviewers
suggestion the result section has now been adapted to include the following sentence: "analyses were
performed in the genome-wide complex trait analysis (GCTA) software and the analyses for 'lipids' and
'organic acids' used unique class-specific and non-class genetic relationship matrices (GRMs) created in
LDAK."

*12 - In the abstract, I think it would be better to state the absolute values, then relative values - atm, you*
*state abs h2total, then relative h2gwas & relative h2snp, then abs h2snp & h2gwas*

We thank the reviewer for the insights into improving the readability of the abstract by including only the
absolute values and removing the relative values. As mentioned in the reply to comment 6, due to the use
of unconstrained variances in the current version of the manuscript, many of the heritability estimates,
especially for h^2_{SNP} , are now negative. Therefore, the ratios (relative values) as defined by Witte et al.
(2014) cannot be calculated for some of our metabolites or result in uninterpretable estimates for other
metabolites. For these reasons we have removed all references to these ratios not only from the current
abstract, but also from the entire manuscript.

**Reviewer #2:**

*1. The authors should change from using "metabolites" to something that describes all 357 measures*
*better, such as "metabolic measures". For example, if one considers the Nightingale platform, many of the*
*measures are not exactly metabolites (for example "Ratio ApoA1/ApoB").*

The reviewer is correct in pointing out that the measures we label as ‘metabolites’ would be better
described as ‘metabolic measures’. As such we have included the following statement to the introduction
of the manuscript “[...] fasting blood metabolic measures (referred to as metabolites in the remainder of
the text for brevity) [...]”

*2. The abstract requires further polishing. It is rather difficult to follow.*

We thank reviewer 2 for the assessment with regards to the readability and understandability of the
abstract. The entire abstract has been rewritten and expanded with ~50 words.

*3. Symbols for indicating different heritability's are usually in italic, but often not. Please be concise.*

We thank the reviewer for pointing out this oversight. We have made a thorough inspection of the
manuscript to ensure all mentions of the heritability estimates have been properly formatted and placed
in italic.

*4. I find the last part of the paper that deals with selection problematic. In most cases the causal variant or*
*haplotype is not known. Therefore the conclusions drawn from the allele frequency and effect estimate is*
*based on unknown linkage disequilibrium between the associated variant and the true causal allele and*
*because that relationship is unknown, any conclusions drawn from the MAF and beta can be, and most*
*likely are, erroneous. Unless the authors can convincingly justify these analyses, they must be removed*
*from the paper.*

We would like to thank the reviewer with their honest opinion and assessment of the BayesS method. In
their paper introducing the BayesS method Zeng et al. (2018) acknowledge that not including the causal
variants in the GRM will cause the absolute value for selection, S , to be slightly underestimated due to
imperfect tagging. However, as reviewer 1 points out in comment 2 our choice to only include SNPs
previously associated with metabolites, as opposed to genome-wide SNPs, leads to further bias in the
analyses. In addition, we've noticed that the correlation among the submitted BayesS results and the

results as obtained with a larger dataset (based on all published metabolite-SNP associations since 2008,
as described in response to comment 1d of reviewer 1) are lower than anticipated ($r = 0.59$; $p = 1.315e-$
09). We cannot properly address or explain these issues at this time, as such we agree with reviewer 2's
opinion and have removed the BayesS analyses from the manuscript.

*5. How is the GW-loci GRM affected by LD between the SNPs that were used to build it? Many of the SNPs*
*are from the same genetic locus (Figure 1) and can describe the same association signal but just have*
*another metabolic measure associated with it. If you do LD pruning on those 497 SNPs, how many are left*
*in if you use r-squared cutoff of 0.1? Does it affect the results?*

We highly appreciate the suggestion of LD pruning the GRM including the metabolite loci. As mentioned in
response to comments 1d and 1f from reviewer 1, we have drastically altered our GRM design in the
current version of the manuscript.). For each HMDB super class represented in our metabolite data (5, but
only 2 analyzed due to insufficient SNPs in the GRM) we created GRMs capturing solely the metabolite loci
for metabolites of this specific class (henceforth we will refer to the heritability estimates derived from
these GRMs as $h^2_{GW-Class}$). Metabolite loci for each class were identified by including all published
metabolite-SNP associations of the relevant class in a clumping procedure ($r^2 = 0.1$, radius = 500kb). The
lead SNPs identified by the clumping procedure were then used as input to create LD-corrected GRMs in
LDAK. To address whether the inclusion of associations for metabolites of a different biochemical class
lead to biasing of the $h^2_{GW-loci}$ estimates as included in the submitted manuscript, we used the same
technique to create 'non-class' GRMs, which used included the lead SNPs of all metabolite-SNP
associations, excluding the relevant class loci and LD proxies (henceforth we will refer to the heritability
estimates derived from these GRMs as $h^2_{GW-Notclass}$). This resulted in 4-variance component models, as

resolved in the GCTA software. In addition, to further address the LD among the class and non-class GRMs
with the GRM used to estimate h^2_{SNP} , we removed the lead SNPs, their LD-proxies and 50kb surrounding
each SNP from both GRMs from the GRM used to estimate the h^2_{SNP} .

To address the question does LD pruning affect the results, we have repeated the 3-variance
component models with the updated metabolite-SNP associations, and applied the clumping algorithm as
described above to this dataset. This resulted in an LDAK 'hit' GRM including 1,230 SNPs. The correlation
between the original $h^2_{GW-loci}$ estimates and these LD-corrected $h^2_{GW-loci}$ estimates is 0.93 ($p < 2.2e-16$),
indicating the results were only minimally effected by not accounting for LD among the 497 SNPs.

**Data availability**

At behest of the editor we have included the statement about the data availability as included in the
manuscript in this point-by-point letter accompanying the revisions:

"The curated list of all published metabolite-SNP associations is included in **Supplementary Data 1**
and is publicly available through the BBMRI – omics atlas
(<http://bbmri.researchlumc.nl/atlas/#data>). All information on the metabolites in this study are in
**Supplementary Table 1**; with full summary statistics for the four-variance component models
included in **Supplementary Table 2**. The Nightingale Health metabolomics data may be requested
through BBMRI-NL (<https://www.bbmri.nl/Omics-metabolomics>). All (other) data may be accessed,
upon approval of the data access committee, through the Netherlands Twin Register
(ntr.fgb@vu.nl). A reporting summary for this Article is available as Supplementary Information file."

Reviewers' Comments:

Reviewer #1:

Remarks to the Author:

Here are my responses to the comments in turn, followed by a quick summary:

=====

1a - thank you for switching to unconstrained variances, and I'm pleased this has increased the convergence rate. I disagree with your reason for not reexamining quality control: "However, we felt adjusting the quality control filters based on observed results 26 in the same study would be 'bad practice'." Were someone to perform a GWAS and find 1000 (almost certainly) false positives, it would not be considered bad practice to go back and reexamine the quality control.

1b - your method for obtaining SDs is acceptable, thank you

1c - you have explained why you would prefer not to follow my suggestion - I see some justification, but in my view it remains that SNP heritabilities, essentially by definition, should be obtained from unrelated samples, and so at the least a verification that results are consistent if you used only unrelated individuals is (in my opinion) required

1d and 1e - I disagree with your reason for not preferring to estimate the variance explained by associations for any metabolite. I remain of the opinion that h2GWAS is, by definition, the variance explained by gwide significant hits - so these must be specific to each metabolite. Therefore, either you perform a gwas for each metabolite, curate a list of metabolite specific associations from previous literature, or change the aim of your work (ie do not attempt to report h2gwas, but some other less standard statistic, h2anymetabolite (but Im not sure the benefit of this statistic). I appreciate the effort of the alternative analysis you performed, but I can't work out what problem this is addressing (but I'm sorry if this is my misunderstanding)

1f - thank you for the extra analysis, I believe that satisfies my original concern.

1g - thank you for the analysis, I believe that suffices

1h - thank you for sds

2 - authors have removed bayes s from the paper - that solves my concern :)

My minor and very minor comments were addressed.

#####

In my view, the estimates of (total) heritability are interesting, but I believe already reported (albeit in less detail). Estimates of h2snp, and how they vary by class, would be interesting, but for reasons explained above, I'm not convinced by the accuracy. Likewise, estimates of h2gwas would be interesting, but I don't feel these are what are reported. Furthermore, due to the nature of the data (ie not by fault of the authors), I note the sds of estimates appear high to the point results might not be useful (e.g., the average h2snp for lipids reported is 0.05, with a variance 0.24 - that means a confidence interval of -.45 to .55, while the c.i. for the reported h2gwas is 0-0.12)

Signed Doug Speed

Reviewer #2:

Remarks to the Author:

I am happy with the modifications done, I have no further comments.

Reviewer #3:

Remarks to the Author:

General Comments:

General point: While you're estimating the "total" heritability based on SNP data, the 'h²_{total}' is not actually just heritable genetic variation. It includes any environmental variation shared by close relatives, which is likely substantial. Traditional twin models would have h² and c² terms, and your model is combining some of the c² into the h²_{total}, because close relatives share environments to some degree (which likely covaries to some degree with pedigree relatedness), and you haven't accounted for those shared environment elsewhere in the model. Total variance explained would be a better way to treat the term, and to be very clear throughout the manuscript (with appropriate limitations listed in the discussion) of what that estimate is and what it is not. See Zaitlen 2013 PLoS Genetics for more details on that point. Related to that, showing the distributions of the relatedness estimates would help. I see that a previous reviewer commented on the data structure, asking about twins vs. other relatives, and I think a supplementary figure would be helpful for that. I don't see the application of the Zaitlen thresholded GRM method as problematic, but I do think that there should be a full and accurate description of the sources of variation that can contribute to the variance estimated by the thresholded GRM, which are not just genetic, and which clearly have bearing on your conclusions.

Another limitation with respect to the GW loci would clearly be the power of the underlying 40 studies where the curated list of SNPs was taken from. These had very different sample sizes, ranging 200-11K+. Different metabolites were presumably studied in different papers looking at Supp. Table 1. How might differential power influence the resulting cross-class comparisons in Figures 2 & 3? This is not to say that the current study is flawed, because I think it's a great question to ask how much of the variance the previously-identified GW significant loci explain, but it's clearly a structural component that would factor into the current analyses. In general, more detail on what other factors, from sample size to shared environments to other things, might also be influencing the estimates and what limitations those might have in terms of the general conclusions of the current manuscript is needed.

Tables A & B of your rebuttal should be included in the supplements, with a description of why you did not use metabolite-specific GRMs, which was a question I originally had in reading the manuscript until I read through the previous rebuttals. I think most readers would have the same question.

Regarding the disagreement between using previously-identified GW significant loci or running the GWAS within the current sample, I think using previously-identified loci is perfectly fine. The question could be made a bit clearer in the text, that you are estimating the variance explained by previously identified GWAS hits, and in an independent sample trying to gauge how important they are. The fact that they are identified in separate sample actually is better, as it takes care of possible 'winner's curse' issues with in-sample significant hits (which given the small sample size would be unlikely anyhow). The first reviewer's concern about definitions of genome-wide significant hits being metabolite-specific is certainly valid, but I think it's a bit tangential to what I see as the point of that aspect of the analysis, which is to evaluate how much genetic variance previously-identified metabolite associated loci are explaining. Perhaps simply calling the h²_{GWAS} something else, such as h²_{metabolite-hits}, and being very clear of the goal and purpose of that aspect of the analysis in the manuscript, would address the issue. (I'd like to see a clearer interpretation of that piece anyhow.)

LDAK weighted GRMs - see comments below. From the rebuttal, it appears the authors have already substantially changed the analyses at least once, and I realize that there would be substantial work involved in changing or demonstrating the utility of using the LDAK GRMs, which I am sympathetic to. However, the logic of using weighted GRMs using previously clumped SNPs is confusing to me, which seems to be doing the same thing twice and likely removes real signal anyhow. Additional simulations could potentially address these concerns, however.

Detailed Comments:

L119: Focusing on the 369 lipids or 'organic acids and derivatives' seems fine, except that of the 402 metabolites classified in L116-117, there are 336 'lipids' and 53 'organic acids' and no 'organic acids and derivatives'. Which set of metabolites did you exactly use? (336+53=389, so it can't be just the lipids + organic acids.) I'm probably missing something, but even so, it should be clear which classes you used in this text as well as Supp. Table 1.

L270-273: If you have measures for the same individuals on different platforms, please also provide a measure of reliability or concordance across those measurements.

L349-357: If you had already imputed all individuals to GONL, why also then impute those (GONL-imputed data) to 1KG on the Michigan server? What was gained from that? Please explain why you imputed twice, and what the benefit of imputing to GONL and then using those imputed data to impute again as opposed to simply using the array data to impute to 1KG to begin with.

L389-390: For GCTA-style analyses of all the G-W significant SNPs simultaneously in one GRM, clumping is unnecessary, and in fact will remove any additional signal that could be gained from additional SNPs with small effects that are in partial LD with the lead SNP. The 'GCTA' model for GRMs can actually account for the LD among these SNPs (see the Yang 2015 Nat. Gen. supplemental derivations). Please clarify or explain in more detail.

L397: LDAK weights the markers by their LD, which you've essentially already done by clumping. What is to be gained by then weighting them somehow? Further, were the LD weights based on the clumped (pruned) set of SNPs or were the weights calculated before the clumping was performed? Again, this will change the weights, because after clumping you've removed much of the LD of the markers in the set, which will increase their weights in the LDAK matrix, and probably change the GRM drastically. With only 500 GW SNPs in the GRM, which are presumably spread across the genome, that were already clumped, I imagine the weighted (LDAK) and unweighted (GCTA) GRMs would be quite similar anyhow (this is certainly not the case for the main weighted GRM, which includes all the imputed variants outside of GW loci, but perhaps for the matrix with only 500 clumped variants, they're largely identical). In general, the logic and reasoning for using the clumping and weighted GRMs isn't clear, and as noted below, the simulations don't really address the issue.

L399-: Did you remove the GW associated SNPs before calculating the G1 and G2 matrices? It's not clear if there was one set of G1 and G2 matrices for each of the metabolites or a single pair of G1 and G2 matrices used for every metabolite, but you wouldn't want to include the GW associated SNPs for a metabolite in the G1 and G2 matrices as well. (I see in L148 of the rebuttal that you removed lead SNPs and those within 50Kb of the lead SNPs from the matrix, which takes care of my concern. That wasn't clear, however, from the methods. Also, the rebuttal paragraph beginning on L362 discusses three variance component models, which I don't see anywhere in the main text or methods or supplementals. What is this analysis, how was it performed, and where are the results?)

L406: LDAK estimates of relatedness will be altered by the weights. Using those within a thresholded matrix (the V(G2) estimate) is likely not the same thing as thresholding based on an unadjusted GRM (GCTA-based), or if it is close, you should demonstrate that the same pairs are

set to 0 for each approach. It would be simple to compare the two matrices and check to see if the thresholded (set to 0) pairs are the same, and if they are, then that's fine, but if the thresholded pairs set to 0 are different, then you will have to explore how that changes the estimates, particularly since the Zaitlen et al. paper, and all others I'm aware of using close relatives, have used an unweighted GRM for that purpose.

L420: How did you compare the models? Supplementary note 3 doesn't describe any formal comparisons. Log likelihood? AIC? BIC? Something else? As the standard errors are likely to be quite large for the h^2 SNP estimate, just finding similar point estimates doesn't seem appropriate or sufficient.

L432: What exactly did you resample? How did you perform your random sampling? Why not just use the standard approach for this, the delta method? (Yang 2015 applied that approach, and it is also in the back of the classic Lynch and Walsh "Quantitative Genetics" textbook.)

Supplementary Note 2: What were the underlying data used for the simulations? Array SNPs, imputed variants, or WGS? That makes a huge amount of difference with these kinds of simulations, because the array and imputed variants are not a random subset of all WGS variants, but rather are better-tagged and are more common than average (and likely more so than unknown causal variants in real data). Using imputed or array data as the underlying simulation leads to a situation where the LD is baked into the simulations already, matching the LDAK model, so of course it will look 'better' but simply because of the LD relationships already in the data. The scenario of randomly chosen SNPs is also quite different from your strategy of first clumping variants with plink before making the matrix, because in the real data, you've forced all the markers in the matrix to be in low LD with one another. With imputed or array variants used for causal variants in your simulations, the simulations are going to show LD-dependent models working better, though that may not have much relevance to real data where the causal variants are unknown, and likely not array SNPs or well-tagged imputed variants. Additional simulations to establish that thresholding the weighted GRM rather than an unweighted GRM are unbiased would be good to see to address the concern of L406 above.

Point-by-point response to the reviewer comments:

The genomic architecture of blood metabolites based on a decade of genome-wide analyses

Fiona A. Hagenbeek^{1,2*}, René Pool^{1,2}, Jenny van Dongen^{1,2}, Harmen H.M. Draisma¹, Jouke Jan Hottenga¹,
Gonneke Willemsen¹, Abdel Abdellaoui¹, Iryna O. Fedko¹, Anouk den Braber^{1,3}, Pieter Jelle Visser^{3,4}, Eco
J.C.N. de Geus^{1,2,5}, Ko Willems van Dijk⁶, Aswin Verhoeven⁷, H. Eka Suchiman⁸, Marian Beekman⁸, P. Eline
Slagboom⁸, Cornelia M. van Duijn⁹, BBMRI Metabolomics Consortium¹⁰, Amy C. Harms¹¹, Thomas
Hankemeier¹¹, Meike Bartels^{1,2,5}, Michel G. Nivard^{1,2,5*‡} and Dorret I. Boomsma^{1,2,5*‡}

**Reviewer #1**

We very warmly thank the reviewer for taking the time to critically appraise our manuscript once more.

We are pleased to have addressed major concerns 1b, 1f-h and 2, as well as the minor and very minor
comments to the satisfaction of reviewer #1. Below we address remaining concerns 1a and 1c-e.

*1a - thank you for switching to unconstrained variances, and I'm pleased this has increased the*
*convergence rate. I disagree with your reason for not reexamining quality control: "However, we felt*
*adjusting the quality control filters based on observed results in the same study would be 'bad practice'."*
*Were someone to perform a GWAS and find 1000 (almost certainly) false positives, it would not be*
*considered bad practice to go back and reexamine the quality control.*

Upon request of reviewer #1, we reexamined the QC of our metabolomics data, which consisted of the
following steps: 1) exclude metabolites with mean coefficient of variation larger than 25%; 2) exclude
metabolites with more than 5% missing data; 3) impute metabolite values below the limit of
detection/quantification with half of the limit; if this was unavailable with half of the lowest observed
value; 4) impute values for outliers (more than 5 SD removed from the mean) with 'mice'; and 5) apply
transformation: inverse normal rank transformation for metabolites measured on NMR platforms and
natural logarithm transformation for metabolites measured on mass spectrometry (MS) platforms.

With these criteria we obtained a convergence rate of nearly 100% (97.8%.) Of the 8 (2.2%) metabolites
that failed to converge, 2 were measured on an MS-platform (e.g., log-normal transformed), and 6 on an
NMR platform (e.g., inverse rank normal transformed). Therefore, non-normality of the phenotype is
unlikely the driver behind non-convergence. Next, we plotted the residuals (after regression out the
covariates as included in the GCTA analyses) of the 6 metabolites as analyzed on an NMR platform. As can
be seen in **Figure A** there are deviations from normality for these 6 metabolites. Plotting the residuals
from 6 random metabolites of the same platform also shows deviations from normality, but less severe
(**Figure B**). Together this suggests non-normality of the residuals as an explanation of non-convergence.

**Figure A.** Density and QQ-plot of the residuals of the 6 rank-inverse normal transformed metabolites
which did not converge in GCTA. **(a)** Density plot of Citrate. **(b)** QQ-plot of Citrate. **(c)** Density plot of PC.
**(d)** QQ-plot of PC. **(e)** Density plot of SM. **(f)** QQ-plot of SM. **(g)** Density plot of S-HDL-FC. **(h)** QQ-plot of S-
HDL-FC. **(i)** Density plot of XS-VLDL-C. **(j)** QQ-plot of XS-VLDL-C. **(k)** Density plot of XS-VLDL-CE. **(l)** QQ-plot
of XS-VLDL-CE.

**Figure B.** Density and QQ-plot of the residuals of the 6 rank-inverse normal transformed metabolites
 which did not converge in GCTA. **(a)** Density plot of Isoleucine. **(b)** QQ-plot of Isoleucine. **(c)** Density plot of
 Leucine. **(d)** QQ-plot of Leucine. **(e)** Density plot of IDL-C. **(f)** QQ-plot of IDL-C. **(g)** Density plot of LDL-TG.
 **(h)** QQ-plot of LDL-TG. **(i)** Density plot of S-HDL-C. **(j)** QQ-plot of S-HDL-C. **(k)** Density plot of XL-HDL-P. **(l)**
 QQ-plot of XL-HDL-P.

*1c - you have explained why you would prefer not to follow my suggestion - I see some justification, but in*
 *my view it remains that SNP heritabilities, essentially by definition, should be obtained from unrelated*
 *samples, and so at the least a verification that results are consistent if you used only unrelated individuals*
 *is (in my opinion) required.*

The paper by Zaitlen et al. (2013) introduced the simultaneous estimation of total and SNP heritability, as
 adopted in the current manuscript. The 2013 paper demonstrated that these two variance components
 can be simultaneously estimated with $h^2_{g,IBS+}$ to estimate h^2_g (genetic variance due to genotyped SNPs) and
 $h^2_{IBS>t+}$ to estimate h^2 (total narrow-sense heritability). The Zaitlen et al. paper (2013) has been cited 137
 46 times according to Web of Science, 126 of these citations came from research articles (**Figure Ca**), many
 were published in high impact journals (**Figure Cb**) and many different disciplines have contributed to the
 citations of the Zaitlen et al. (2013) paper (**Figure Cc**). The threshold GRM method of Zaitlen et al. (2013)

has become widely used when including related and unrelated samples in GREML analyses, particularly for
 twin-family cohorts. We greatly value the comments of the reviewer, and recognize that more methods
 comparisons are possible. However, we would prefer our paper to focus on a decade of metabolomics
 GWASs.

**Figure C.** Web of Science citation analysis for the Zaitlen et al. (2013) paper detailing the threshold GRM
 method. **(a)** Overview of the type of articles in Web of Science citing the Zaitlen et al. paper. **(b)** Overview
 of the top 20 journals publishing papers citing the Zaitlen et al. paper in Web of Science. **(c)** Overview of
 the top 10 disciplines of the papers citing the Zaitlen et al. paper in Web of Science.

*1d and 1e - I disagree with your reason for not preferring to estimate the variance explained by*
*associations for any metabolite. I remain of the opinion that h2GWAS is, by definition, the variance*
*explained by gwide significant hits - so these must be specific to each metabolite. Therefore, either you*
*perform a gwas for each metabolite, curate a list of metabolite specific associations from previous*
*literature, or change the aim of your work (i.e. do not attempt to report h2gwas, but some other less*
*standard statistic, h2anymetabolite (but I'm not sure the benefit of this statistic). I appreciate the effort of*
*the alternative analysis you performed, but I can't work out what problem this is addressing (but I'm sorry*
*if this is my misunderstanding).*

Thank you for your detailed comments and suggestions. We have followed the suggestion that we
redefine the aim of our study. We hypothesize that the genetic variance of metabolite loci of the not-
target super classes reflect genetic signals that current GWA and (exome-) sequencing studies for the
target metabolite have so far been underpowered to detect. To distinguish our aim, from the aim of
evaluating how much of the genetic variance in a metabolite is due to the metabolite-specific loci, we no
longer refer to the sum of the super class-specific and other super classes variance as $h^2_{GW-loci}$, but as
$h^2_{metabolite-hits}$ (in line with the suggestion of reviewer #3). We've made the necessary adjustments to the
manuscript to better reflect the aim of our analyses: **Introduction** (L84-89), and the **Results** (L134-155).

Finally, we must address reviewer #1's concern that we are ignoring the request to investigate the genetic
variance of only the metabolite-specific loci. We actively argued against using metabolite-specific GRMs to
estimate the genetic variance of metabolite-specific loci in our previous rebuttal. There, we attempted to
meet reviewer #1 half way: instead of using a single GRM with all metabolite loci we first attempted to
create several GRMs using class-specific GRMs. However, as shown in **Table A** of the previous rebuttal
(now included as **Supplementary Table 2**), creating GRMs for the specific classes (with a small number of
SNPs per GRM) resulted in extremely high rates of non-convergence (37.9%). Using super classes instead
of classes resulted in GRMs with a larger number of SNPs and a convergence rate of 97.8% (**Table B** in
previous rebuttal; now included as **Supplementary Table 4**). A description of these efforts have been

included in **Supplementary Note 2** and at the beginning of the **Results** section (L134-148). Creating
metabolite-specific GRMs will result in GRMs with small numbers of SNPs, comparable to, or smaller than,
the number of SNPs included in the class-specific GRMs. Therefore, convergence rates for analyses
including only metabolite-specific GRMs will be as poor, if not poorer than those observed for the class-
specific GRMs.

*In my view, the estimates of (total) heritability are interesting, but I believe already reported (albeit in less*
*detail). Estimates of h^2_{SNP} , and how they vary by class, would be interesting, but for reasons explained*
*above, I'm not convinced by the accuracy. Likewise, estimates of h^2_{Gwas} would be interesting, but I don't*
*feel these are what are reported. Furthermore, due to the nature of the data (i.e. not by fault of the*
*authors), I note the SD's of estimates appear high to the point results might not be useful (e.g., the average*
*h^2_{SNP} for lipids reported is 0.05, with a variance 0.24 - that means a confidence interval of -.45 to .55,*
*while the c.i. for the reported h^2_{Gwas} is 0-0.12).*

Thank you for this comments. We agree with the observation that the SE's for h^2_{SNP} estimates are high,
resulting in non-informative h^2_{SNP} estimates. We moved the h^2_{SNP} estimates to the supplements, as such
the h^2_{SNP} estimates no longer play a large role in the current version of the manuscript.

**Reviewer #2**

We would like to thank the reviewer for appraising our manuscript and rebuttal and are happy we've been
able to modify the previous version of the manuscript to the reviewer's specifications. We trust that the
changes to the current version of the manuscript are still in line with reviewer #2's expectations.

**Reviewer #3**

**General Comments:**

*General point: While you're estimating the "total" heritability based on SNP data, the 'h²_{total}' is not*
*actually just heritable genetic variation. It includes any environmental variation shared by close relatives,*
*which is likely substantial. Traditional twin models would have h² and c² terms, and your model is*
*combining some of the c² into the h²_{total}, because close relatives share environments to some degree*
*(which likely covaries to some degree with pedigree relatedness), and you haven't accounted for those*
*shared environment elsewhere in the model. Total variance explained would be a better way to treat the*
*term, and to be very clear throughout the manuscript (with appropriate limitations listed in the discussion)*
*of what that estimate is and what it is not. See Zaitlen 2013 PLoS Genetics for more details on that point.*
*Related to that, showing the distributions of the relatedness estimates would help. I see that a previous*
*reviewer commented on the data structure, asking about twins vs. other relatives, and I think a*
*supplementary figure would be helpful for that. I don't see the application of the Zaitlen thresholded GRM*
*method as problematic, but I do think that there should be a full and accurate description of the sources of*
*variation that can contribute to the variance estimated by the thresholded GRM, which are not just*
*genetic, and which clearly have bearing on your conclusions.*

We are very grateful to reviewer #3 for reviewing our manuscript. The reviewer points out that our h^2_{total}
has not been properly defined in the previous version of the manuscript and that the definition and
limitations of h^2_{total} should be addressed in more detail in the **Discussion** section. As reviewer #3 notes, our
h^2_{total} estimates may contain effects of shared environment, dominance and epistasis when including large
proportions of closely related individuals. We have nuanced the definition of h^2_{total} in the **Introduction**
(L76-78, the **Discussion** (L258-277), and the **Methods** (L492-495). However, while our sample does contain
closely related individuals, our sample is more complex which likely attenuates the potential bias
(**Supplementary Figure 3**; included per the reviewers recommendation). As requested, on L255-290 in the

discussion we have addressed the limitations of a threshold-GRM design, including the additional sources
of variation contributing to the total variance.

In the limitation section of the **Discussion** we describe that the effects of the shared environment,
dominance and epistasis have not been well studied for metabolite levels. For example, while many twin-
family studies have investigated ‘familiality’ (combination of genetic and common environmental effects),
only a small number of studies explicitly report effect of common environment. Common environmental
influences are reported for a small number of metabolites (e.g., 14.3% of all Nightingale Health ¹H-NMR
metabolites) and the influence of the shared environment is small-to-moderate (platform and metabolite
class-dependent means range from 0.03 to 0.45). Below, we offer a description of studies reporting the
effect of the common environment for platforms similar or equal to those used in our own study:

Nightingale Health NMR (i.e., NMR with lipoprotein fractions): Kettunen et al 2012 (DOI:10.1038/ng.1073)
estimated ACE or ADE models of 217 metabolites of the Nightingale Health NMR platform, targeting
lipoprotein fractions, amino acids and ratios thereof. Kettunen et al (2012) found that the best model was
an AE model for 78.8%, an ACE model for 11.5%, an ADE model for 6.45% and an CE model for 2.76% of
the metabolites. Thus, shared environment is included in the model for 14.28% of the metabolites and
explains on average 3.2% (range: 0.00-0.38) of the variance in metabolite levels for the Nightingale Health
NMR platform.

Biocrates (i.e., MS with lipids, acyl carnitines and amino acids): Menni et al 2013 (DOI 10.1007/s11306-
012-0469-6) observed an effect for the shared environmental influences for only a small number of
Biocrates metabolites, with an average c^2 of 3.8%. Yet et al 2016 (DOI:10.1371/journal.pone.0153672)
report heritability estimates for 43 metabolites included on both the Biocrates and the Metabolon
platforms. In their study they report an average c^2 of 0.16 for metabolites of the Biocrates platform and of
0.11 for the same metabolites on the Metabolon platform; platform has a small influence on the genetic
etiology of metabolite levels, though this might be attributable to differences in measurement noise
between platforms. A parent-offspring study reported c^2 estimates of 0% to 58.1%, with an average of

6.2% for metabolites of the Biocrates platform (Tremblay et al. 2019; DOI: 10.1016/j.nutres.2018.10.003).
In a subset of NTR Biocrates data, Draisma et al. 2013 (DOI:10.1017/thg.2013.59) reported a mean spousal
correlation of 0.24 which reflects either assortative mating or household sharing.

Lipidomics: Frahnnow et al. 2017 (DOI:10.1038/s41598-017-03965-6) investigated the genetic etiology of a
lipidomics dataset by lipid subclasses. Here, they reported an overall effect of c^2 of approximately 45%
across all lipids, where no or only small effects of c^2 were observed for sphingomyelins,
phosphatidylethanolamines, lyso-phosphatidylcholines, triglycerides and lyso-phosphatidylethanolamines.
In contrast, large effects of c^2 , with small to no contributions of a^2 , were reported for sterols,
phosphatidylinositols, phosphatidylcholines, diacylglycerols, ceramides, phosphatidylethanolamine ethers,
phosphatidylcholine ethers and sterolesters.

**Supplementary Figure 3.** Histogram of the off-diagonal elements of the Genetic Relatedness Matrix (GRM)
of all participants. The off-diagonal elements represent the pairwise relatedness for all participants. **(a)**
Histogram of all off-diagonal elements of the LDAK GRM for all participants. As a large proportion of
relationships among the participants is unrelated (<0.05) this histogram is highly skewed to the left,
making the relationships among related individuals undistinguishable. **(b)** Histogram of the off-diagonal
elements in the LDAK GRM for the unrelated (<0.05) participants. **(c)** Histogram of all off-diagonal
elements of the LDAK GRM for all related (>0.05) participants.

*Another limitation with respect to the GW loci would clearly be the power of the underlying 40 studies*
 *where the curated list of SNPs was taken from. These had very different sample sizes, ranging 200-11K+.*
 *Different metabolites were presumably studied in different papers looking at Supp. Table 1. How might*
 *differential power influence the resulting cross-class comparisons in Figures 2 & 3? This is not to say that*
 *the current study is flawed, because I think it's a great question to ask how much of the variance the*
 *previously-identified GW significant loci explain, but it's clearly a structural component that would factor*
 *into the current analyses. In general, more detail on what other factors, from sample size to shared*
 *environments to other things, might also be influencing the estimates and what limitations those might*
 *have in terms of the general conclusions of the current manuscript is needed.*

Thank you for pointing out that “ *it's a great question to ask how much of the variance the previously-*
 *identified GW significant loci explain*”. We also thank reviewer #3 for stressing the point of varying sample
 sizes in the GWAS discovery and have added that our results are conditional on the power of the studies

of the past decade in our **Results** (L134-138), specifically referring to the overview of the different sample
sizes per study as provided in **Supplementary Note 1**. Finally, we added a limitation section to the
**Discussion** (L255-290), where a description is given of the factors that might influence the heritability
estimates, including the varying power of the underlying studies the metabolite loci were extracted from.
*Tables A & B of your rebuttal should be included in the supplements, with a description of why you did not*
*use metabolite-specific GRMs, which was a question I originally had in reading the manuscript until I read*
*through the previous rebuttals. I think most readers would have the same question.*

We very much appreciate the suggestion to include Tables A and B of the rebuttal in the supplements. In
**Supplementary Note 2** we have described our efforts at constructing metabolite class-specific GRMs,
where we show that using the more fine-grained HMDB metabolite classes, as opposed to the HMDB
super-classes, results in high degrees of non-convergence; this likely related to the smaller number of
SNPs included in the GRMs. Tables A and B from the rebuttal have been included as **Supplementary**
**Tables 2** and **4**. A brief summary of **Supplementary Notes 2** has been given on L139-148 in the **Results**.

*Regarding the disagreement between using previously-identified GW significant loci or running the GWAS*
*within the current sample, I think using previously-identified loci is perfectly fine. The question could be*
*made a bit clearer in the text, that you are estimating the variance explained by previously identified*
*GWAS hits, and in an independent sample trying to gauge how important they are. The fact that they are*
*identified in separate sample actually is better, as it takes care of possible ‘winner’s curse’ issues with in-*
*sample significant hits (which given the small sample size would be unlikely anyhow). The first reviewer’s*
*concern about definitions of genome-wide significant hits being metabolite-specific is certainly valid, but I*
*think it’s a bit tangential to what I see as the point of that aspect of the analysis, which is to evaluate how*
*much genetic variance previously-identified metabolite associated loci are explaining. Perhaps simply*
*calling the h2GWAS something else, such as h2metabolite-hits, and being very clear of the goal and*
*purpose of that aspect of the analysis in the manuscript, would address the issue. (I’d like to see a clearer*
*interpretation of that piece anyhow.)*

Thank you for noting that “ *using previously-identified loci is perfectly fine*”. We also appreciate the
suggestion to clarify the aim of our study and to use a different term for $h^2_{GW-loci}$ and have done so. To
distinguish our aim, estimate the genetic variance of previously identified metabolite loci in an
independent sample, from the aim of evaluating how much of the genetic variance in a metabolite is due
to the metabolite-specific loci, we no longer refer to the sum of the super class-specific and other super
classes variance as $h^2_{GW-loci}$, but as $h^2_{metabolite-hits}$ (in line with the suggestion of reviewer #3). We’ve made
the necessary adjustments to the manuscript to better reflect the aim of our analyses: **Introduction** (L84-
89), and the **Results** (L134-155).

*LDAK weighted GRMs - see comments below. From the rebuttal, it appears the authors have already*
*substantially changed the analyses at least once, and I realize that there would be substantial work*
*involved in changing or demonstrating the utility of using the LDAK GRMs, which I am sympathetic to.*
*However, the logic of using weighted GRMs using previously clumped SNPs is confusing to me, which*
*seems to be doing the same thing twice and likely removes real signal anyhow. Additional simulations*
*could potentially address these concerns, however.*

Thank you for recognizing the huge amount work we did. We appreciate the concern with regard to
clumped SNPs in a weighted LDAK GRM. In the first draft of our manuscript we used the weighted LDAK
GRM without prior clumping of the SNPs. However, in the previous round of rebuttals reviewer #2 had the
following concern: “*How is the GW-loci GRM affected by LD between the SNPs that were used to build it?*
*Many of the SNPs are from the same genetic loci and can describe the same association signal but just*
*have another metabolic measure associated with it [..]”. To accommodate reviewer #2 we included*
clumped GRMs in the second draft of the manuscript.

To address the concern of reviewer #3 about overcorrecting a weighted GRM when using clumped SNPs,
and possibly removing real signal, we compared the clumped and not-clumped LDAK GRMs. We used the
GCTA power calculator to compare the estimated SE’s and power for the different GRMs included in our
manuscript using clumped and not-clumped SNPs (<https://cnsgenomics.shinyapps.io/gctaPower/>). The

differences between the GRMs using clumped or not-clumped SNPs are very small (**Supplementary Table**
**16**). Across all 5 GRMs the average mean differences between the clumped and not-clumped GRMs are
negative, indicating that clumped GRMs results in smaller SE's, though with a minimal difference
(**Supplementary Table 16**). Similarly, the average power differences are zero or positive (again with
minimal differences), indicating that clumped GRMs are as powerful as, or more powerful than not-
clumped GRMs (**Supplementary Table 16**). While the gain of using a clumped weighted LDAK GRM versus
a not-clumped LDAK GRM is small, we feel that given the large SE's we observed in our study, the choice
for clumped weighted LDAK GRMs is valid. This description has been included in **Supplementary Note 3**.

**Supplementary Table 16.** Results power analysis, using the GCTA power calculator, to compare the clumped LDAK GRMs with the not-clumped LDAK genetic
 relatedness matrices (GRM). The Type I error rate, α , has been set to 0.05 for all power calculations. We've calculated the power assuming the true SNP
 heritability (h^2_{SNP}) per GRM is 0.20, 0.10 or 0.05. The variance of the SNP-derived genetic relationships is based on the variance of the off-diagonal elements of
 each GRM. The power calculates provides us with the standard error (SE) of h^2_{SNP} , the non-centrality parameter (NCP) of the chi-squared test statistic ($h^4/(SE)^2$)
 and power (probability of detecting $h^2_{SNP} > 0$). Explanations of the different GRMs can be found in the **Methods**, and in **Supplementary Figure 1**.

Type GRM	Platform	h^2_{SNP}	Parameter	$V(G1)$	$V(G3)$ – lipid loci	$V(G4)$ – excl. lipid loci	$V(G3)$ - organic acid loci	$V(G4)$ – excl. organic acid loci	$V(G3)$ all metabolite loci	
Clumped			Variance GRM	0.00012081	0.002402	0.0019347	0.00285407	0.0017119	0.00092593	
	Nightingale Health N = 4,227	0.2	SE	0.0304	0.0068	0.0076	0.0063	0.0081	0.011	
			NCP	43.1715	858.3601	691.3662	1019.9036	611.7485	330.8816	
			Power	1	1	1	1	1	1	
	Lipidomics & Leiden-NMR N = 2,324	0.2	SE	0.0554	0.0124	0.0138	0.0114	0.0147	0.0199	
			NCP	13.0498	259.4478	208.9854	308.2953	184.9186	100.0185	
			Power	0.9508	1	1	1	1	1	
	Biocrates N = 1,448	0.2	SE	0.0889	0.0199	0.0222	0.0183	0.0236	0.0321	
			NCP	5.0661	100.7198	81.1299	119.6828	71.787	38.828	
			Power	0.6144	1	1	1	1	1	
	Nightingale Health N = 4,227	0.1	SE	0.0304	0.0068	0.0076	0.0063	0.0081	0.011	
			NCP	10.7929	214.5765	172.8415	254.9759	152.9371	82.7204	
			Power	0.9075	1	1	1	1	1	
	Lipidomics & Leiden-NMR N = 2,324	0.1	SE	0.0551	0.0124	0.0138	0.0113	0.0146	0.0199	
			NCP	3.2906	65.4213	52.6969	77.7385	46.6284	25.2203	
			Power	0.4421	1	1	1	1	0.9989	
	Biocrates			SE	0.0889	0.0199	0.0222	0.0183	0.0236	0.0321

Type GRM	Platform	$h^2_{s_{NP}}$	Parameter	$V(G1)$	$V(G3) - \text{lipid loci}$	$V(G4) - \text{excl. lipid loci}$	$V(G3) - \text{organic acid loci}$	$(VG4) - \text{excl. organic acid loci}$	$V(G3) \text{ all metabolite loci}$
Not Clumped	N = 1,448	0.05	NCP	1.2665	25.1799	5.0706	29.9207	17.9467	9.707
			Power	0.203	0.9989	0.7281	0.998	0.9886	0.8761
	Nightingale Health N = 4,227		SE	0.0304	0.0068	0.0076	0.0063	0.0081	0.011
			NCP	2.6982	53.6441	43.2104	63.744	38.2343	20.6801
			Power	0.3757	1	1	1	1	0.9952
	Lipidomics & Leiden-NMR N = 2,324		SE	0.0551	0.0124	0.0138	0.0113	0.0146	0.0199
			NCP	0.8226	16.3553	13.1742	19.4346	11.6571	6.3051
			Power	0.1483	0.9814	0.9525	0.9928	0.9271	0.7092
	Biocrates N = 1,448		SE	0.0889	0.0199	0.0222	0.0183	0.0236	0.0321
			NCP	0.3166	6.295	5.0706	7.4802	4.4867	2.4268
			Power	0.087	0.7085	0.6148	0.8622	0.5629	0.334
			Variance GRM	0.00012022	0.001652	0.00125415	0.00202285	0.00117437	0.00062578
	Nightingale Health N = 4,227		SE	0.0305	0.0082	0.0094	0.0074	0.0098	0.0134
			NCP	42.9607	590.2824	448.1712	722.8666	419.6618	223.6228
			Power	1	1	1	1	1	1
Lipidomics & Leiden-NMR N = 2,324	SE	0.0555	0.015	0.172	0.0135	0.0178	0.0243		
	NCP	12.9861	178.4299	135.4727	218.5073	126.8549	67.5965		
	Power	0.9499	1	1	1	1	1		
Biocrates N = 1,448	SE	0.0891	0.024	0.0276	0.0217	0.0285	0.039		
	NCP	5.0413	69.268	52.5916	84.8264	49.2461	26.2415		
	Power	0.6123	1	1	1	1	0.9992		
Nightingale Health N = 4,227	SE	0.0305	0.0082	0.0094	0.0074	0.0098	0.0134		
	NCP	10.7402	147.5706	112.0428	180.7167	104.9155	55.9057		
	Power	0.9061	1	1	1	1	1		
Lipidomics & Leiden-NMR	SE	0.0555	0.015	0.0172	0.0135	0.0178	0.0243		
	NCP	3.2465	44.6075	33.8682	54.6268	31.7137	16.8991		

Type GRM	Platform	$h^2_{s_{NP}}$	Parameter	$V(G1)$	$V(G3) - \text{lipid loci}$	$V(G4) - \text{excl. lipid loci}$	$V(G3) - \text{organic acid loci}$	$(VG4) - \text{excl. organic acid loci}$	$V(G3) \text{ all metabolite loci}$
	N = 2,324		Power	0.4373	1	0.9999	1	0.9999	0.9843
	Biocrates		SE	0.0891	0.024	0.0276	0.0217	0.0285	0.039
	N = 1,448		NCP	1.2603	17.317	13.1479	21.2066	12.3115	6.5604
			Power	0.2022	0.9861	0.9521	0.9959	0.9393	0.7262
	Nightingale Health		SE	0.0305	0.0082	0.0094	0.0074	0.0098	0.0134
	N = 4,227		NCP	2.685	36.8927	28.0107	45.1792	26.2289	13.9764
			Power	0.3741	1	0.9996	1	0.9992	0.9623
	Lipidomics & Leiden-NMR	0.05	SE	0.0555	0.015	0.0172	0.0135	0.0178	0.0243
	N = 2,324		NCP	0.8116	11.1519	8.467	13.6567	7.9284	4.2248
			Power	0.1469	0.9161	0.8289	0.9587	0.8039	0.5381
	Biocrates		SE	0.0891	0.024	0.0276	0.0217	0.0285	0.039
	N = 1,448		NCP	0.3151	4.3292	3.287	5.3016	3.0779	1.6401
			Power	0.0868	0.5481	0.4417	0.634	0.4187	0.2491

Detailed Comments:

*L119: Focusing on the 369 lipids or ‘organic acids and derivatives’ seems fine, except that of the 402*
*metabolites classified in L116-117, there are 336 ‘lipids’ and 53 ‘organic acids’ and no ‘organic acids and*
*derivatives’. Which set of metabolites did you exactly use? (336+53=389, so it can’t be just the lipids +*
*organic acids.) I’m probably missing something, but even so, it should be clear which classes you used in*
*this text as well as Supp. Table 1.*

We thank reviewer #3 for noticing the discrepancy between L116-117 and L119. The GCTA analyses were
indeed performed for 369 metabolites, of which 316 were classified as ‘lipids’ and 53 as ‘organic acids’.
The discrepancy between L116-117 and L119 in the previous version of the manuscript is due to the fact
that we neglected to add to L119 that of the 389 metabolites classified as either lipids or organic acids 20
did not pass QC. Therefore, the GCTA analyses included only 369 metabolites as opposed to 389
metabolites. In the current version of the manuscript we have adjusted L119 (now L125-126) to indicate
that this is the number of lipids and organic acids that pass QC.

*L270-273: If you have measures for the same individuals on different platforms, please also provide a*
*measure of reliability or concordance across those measurements.*

The correlation of metabolites included on multiple platforms for overlapping individuals were provided in
**Supplementary Table 3** in the previous version of the manuscript. In the current version of the
manuscript, this has now been referred to as **Supplementary Table 5** and we have explicitly mentioned
this on L179.

*L349-357: If you had already imputed all individuals to GONL, why also then impute those (GONL-imputed*
*data) to 1KG on the Michigan server? What was gained from that? Please explain why you imputed twice,*
*and what the benefit of imputing to GONL and then using those imputed data to impute again as opposed*
*to simply using the array data to impute to 1KG to begin with.*

Because genotyping in NTR participants has been done on different genotyping arrays throughout the
270 years, the first step before imputation is to create a single cross-imputed dataset with GoNL as the
271 reference. Here we do not impute to all variants included in GoNL, but limit to variants that are present on
at least one genotyping array. This cross-imputed dataset is used when we create GRMs and is used for
polygenic prediction and as backbone for imputation (see e.g., Fedko et al. 2015 [DOI 10.1007/s10519-
015-9725-7], and Abdellaoui et al. 2018 [DOI: 10.1111/gbb.12472]).

*L389-390: For GCTA-style analyses of all the G-W significant SNPs simultaneously in one GRM, clumping is*
*unnecessary, and in fact will remove any additional signal that could be gained from additional SNPs with*
*small effects that are in partial LD with the lead SNP. The 'GCTA' model for GRMs can actually account for*
*the LD among these SNPs (see the Yang 2015 Nat. Gen. supplemental derivations). Please clarify or explain*
*in more detail.*

This concern has been addressed on L224-243 of the current rebuttal.

*L397: LDAK weights the markers by their LD, which you've essentially already done by clumping. What is to*
*be gained by then weighting them somehow? Further, were the LD weights based on the clumped (pruned)*
*set of SNPs or were the weights calculated before the clumping was performed? Again, this will change the*
*weights, because after clumping you've removed much of the LD of the markers in the set, which will*
*increase their weights in the LDAK matrix, and probably change the GRM drastically. With only 500 GW*
*SNPs in the GRM, which are presumably spread across the genome, that were already clumped, I imagine*
*the weighted (LDAK) and unweighted (GCTA) GRMs would be quite similar anyhow (this is certainly not the*
*case for the main weighted GRM, which includes all the imputed variants outside of GW loci, but perhaps*
*for the matrix with only 500 clumped variants, they're largely identical). In general, the logic and reasoning*
*for using the clumping and weighted GRMs isn't clear, and as noted below, the simulations don't really*
*address the issue.*

We thank the reviewer and highly appreciate the remarks about the likely similarities between the
weighted LDAK GRM and GCTA GRM after clumping of the SNPs. The gain of using clumped SNPs to create
the weighted LDAK GRM, as compared to creating a weighted LDAK GRM from not-clumped SNPs is small
as described on L224-243 of the current rebuttal. Additionally, in **Figure D** we provide the distribution of
the off-diagonal elements of the clumped and not-clumped LDAK GRMs, showing that the GRMs are highly
similar. We apologize that the order of clumping versus calculating the weights for LDAK was unclear. In
creating the LDAK GRMS we first performed clumping on the metabolite loci, before calculating the
weights. We have adjusted the relevant section in the **Methods** to better reflect this (L456-458).

Reviewer #3 is completely correct in the assessment that clumped LDAK and clumped GCTA GRMs will be
similar for the $V(G3)$ and $V(G4)$ GRMs with ~ 500 loci (**Supplementary Table 17**). Moreover, the $V(G1)$
GRMs (cross-imputed GRM without metabolite loci) are also similar when based on clumped SNPs created
in either LDAK or GCTA (**Supplementary Table 17**). In fact, we observe slightly lower SE's and slightly
higher power for the clumped LDAK GRM as compared to the clumped GCTA GRM (**Supplementary Table**
**17**). Given that a detailed comparison of the pros and cons of the LDAK and GCTA method for the creation
of GRMs is beyond the scope of the current manuscript, we observe comparable power of clumped LDAK
and GCTA GRMs, and observe that clumped LDAK GRMs are more powerful and precise than not-clumped
LDAK GRMs, we have added a short description of the comparison of the clumped LDAK and GCTA GRMs
to **Supplementary Note 3**.

**Figure D.** Distribution of the off-diagonal elements of the clumped and not-clumped LDAK GRM for all
participants. **(a)** Histogram of all off-diagonal elements of the clumped LDAK GRM for all participants. As a
large proportion of relationships among the participants is unrelated (<0.05) this histogram is highly
skewed to the left, making the relationships among related individuals undistinguishable. **(b)** Histogram of
the off-diagonal elements in the clumped LDAK GRM for the unrelated (<0.05) participants. **(c)** Histogram
of all off-diagonal elements of the clumped LDAK GRM for all related (>0.05) participants. **(d)** Histogram of

all off-diagonal elements of the not-clumped LDAK GRM for all participants. As a large proportion of
relationships among the participants is unrelated (<0.05) this histogram is highly skewed to the left,
making the relationships among related individuals undistinguishable. **(e)** Histogram of the off-diagonal
elements in the not-clumped LDAK GRM for the unrelated (<0.05) participants. **(f)** Histogram of all off-
diagonal elements of the not-clumped LDAK GRM for all related (>0.05) participants. **(g)** scatterplot of all
off-diagonal elements of the clumped LDAK versus not-clumped LDAK GRMs with the relationships for
unrelated individuals (<0.05) set to zero.

**Supplementary Table 17.** Results power analysis, using the GCTA power calculator, to compare the clumped LDAK GRMs with the clumped GCTA genetic
 relatedness matrices (GRM). The Type I error rate, α , has been set to 0.05 for all power calculations. We've calculated the power assuming the true SNP
 heritability (h^2_{SNP}) per GRM is 0.20, 0.10 or 0.05. The variance of the SNP-derived genetic relationships is based on the variance of the off-diagonal elements of
 each GRM. The power calculates provides us with the standard error (SE) of h^2_{SNP} , the non-centrality parameter (NCP) of the chi-squared test statistic ($h^4/(SE)^2$)
 and power (probability of detecting $h^2_{SNP} > 0$). Explanations of the different GRMs can be found in the **Methods**, and in **Supplementary Figure 1**.

Type GRM	Platform	h^2_{SNP}	Parameter	$V(G1)$	$V(G3)$ – lipid loci	$V(G4)$ – excl. lipid loci	$V(G3)$ - organic acid loci	$(VG4)$ – excl. organic acid loci	$V(G3)$ all metabolite loci
Clumped LDAK			Variance GRM	0.00012081	0.00240186	0.0019347	0.00285407	0.0017119	0.00092593
	Nightingale Health N = 4,227		SE	0.0304	0.0068	0.0076	0.0063	0.0081	0.011
			NCP	43.1715	858.3601	691.3662	1019.9036	611.7485	330.8816
			Power	1	1	1	1	1	1
	Lipidomics & Leiden-NMR N = 2,324		SE	0.0554	0.0124	0.0138	0.0114	0.0147	0.0199
		0.2	NCP	13.0498	259.4478	208.9854	308.2953	184.9186	100.0185
			Power	0.9508	1	1	1	1	1
	Biocrates N = 1,448		SE	0.0889	0.0199	0.0222	0.0183	0.0236	0.0321
			NCP	5.0661	100.7198	81.1299	119.6828	71.787	38.828
			Power	0.6144	1	1	1	1	1

Type GRM	Platform	$h^2_{s_{NP}}$	Parameter	$V(G1)$	$V(G3) - \text{lipid loci}$	$V(G4) - \text{excl. lipid loci}$	$V(G3) - \text{organic acid loci}$	$(VG4) - \text{excl. organic acid loci}$	$V(G3) \text{ all metabolite loci}$
	Nightingale Health N = 4,227		SE	0.0304	0.0068	0.0076	0.0063	0.0081	0.011
			NCP	10.7929	214.5765	172.8415	254.9759	152.9371	82.7204
			Power	0.9075	1	1	1	1	1
	Lipidomics & Leiden-NMR N = 2,324	0.1	SE	0.0551	0.0124	0.0138	0.0113	0.0146	0.0199
			NCP	3.2906	65.4213	52.6969	77.7385	46.6284	25.2203
			Power	0.4421	1	1	1	1	0.9989
	Biocrates N = 1,448		SE	0.0889	0.0199	0.0222	0.0183	0.0236	0.0321
			NCP	1.2665	25.1799	5.0706	29.9207	17.9467	9.707
			Power	0.203	0.9989	0.7281	0.998	0.9886	0.8761
	Nightingale Health N = 4,227		SE	0.0304	0.0068	0.0076	0.0063	0.0081	0.011
			NCP	2.6982	53.6441	43.2104	63.744	38.2343	20.6801
			Power	0.3757	1	1	1	1	0.9952
	Lipidomics & Leiden-NMR N = 2,324	0.05	SE	0.0551	0.0124	0.0138	0.0113	0.0146	0.0199
			NCP	0.8226	16.3553	13.1742	19.4346	11.6571	6.3051
			Power	0.1483	0.9814	0.9525	0.9928	0.9271	0.7092
	Biocrates		SE	0.0889	0.0199	0.0222	0.0183	0.0236	0.0321

Type GRM	Platform	$h^2_{s, NP}$	Parameter	$V(G1)$	$V(G3) - \text{lipid loci}$	$V(G4) - \text{excl. lipid loci}$	$V(G3) - \text{organic acid loci}$	$(VG4) - \text{excl. organic acid loci}$	$V(G3) \text{ all metabolite loci}$	
Clumped GCTA	N = 1,448		NCP	0.3166	6.295	5.0706	7.4802	4.4867	2.4268	
			Power	0.087	0.7085	0.6148	0.8622	0.5629	0.334	
				Variance GRM	0.00013276	0.00240547	0.00195309	0.0028819	0.00171796	0.00101533
	Nightingale Health N = 4,227			SE	0.029	0.0068	0.0076	0.0062	0.0081	0.0105
				NCP	47.4419	589.5961	697.9378	1029.8486	613.914	362.8288
				Power	1	1	1	1	1	1
	Lipidomics & Leiden-NMR N = 2,324	0.2		SE	0.0528	0.0124	0.0138	0.0113	0.0147	0.0191
				NCP	14.3407	259.8377	210.9718	311.3015	185.5732	109.6755
				Power	0.9661	1	1	1	1	1
	Biocrates N = 1,448			SE	0.0848	0.0199	0.0221	0.0182	0.0236	0.0307
				NCP	5.5672	100.8712	81.901	120.8498	72.0411	42.5769
				Power	0.6553	1	1	1	1	1
	Nightingale Health N = 4,227		0.1	SE	0.029	0.0068	0.0076	0.0062	0.0081	0.0105
				NCP	11.8605	214.899	174.4854	257.4622	153.4785	90.7072
				Power	0.9311	1	1	1	1	1

Type GRM	Platform	$h^2_{s, NP}$	Parameter	$V(G1)$	$V(G3) - \text{lipid loci}$	$V(G4) - \text{excl. lipid loci}$	$V(G3) - \text{organic acid loci}$	$(VG4) - \text{excl. organic acid loci}$	$V(G3) \text{ all metabolite loci}$
Lipidomics & Leiden-NMR N = 2,324			SE	0.0528	0.0124	0.0138	0.0113	0.0147	0.0191
			NCP	3.5852	64.9594	52.743	77.8254	46.3933	27.4189
			Power	0.4735	1	1	1	1	0.995
Biocrates N = 1,448			SE	0.0848	0.0199	0.0221	0.0182	0.0236	0.0307
			NCP	1.3918	25.2178	20.4753	30.2125	18.0103	10.6442
			Power	0.2185	0.9989	0.9948	0.9998	0.9888	0.9036
Nightingale Health N = 4,227			SE	0.029	0.0068	0.0076	0.0062	0.0081	0.0105
			NCP	2.9651	53.7248	43.6211	64.3655	38.3696	22.6768
			Power	0.4061	1	1	1	1	0.9975
Lipidomics & Leiden-NMR N = 2,324		0.05	SE	0.0528	0.0124	0.0138	0.0113	0.0147	0.019
			NCP	0.8963	16.2399	13.1857	19.4563	11.5983	6.8547
			Power	0.1573	0.9808	0.9527	0.9929	0.9259	0.774
Biocrates N = 1,448			SE	0.0848	0.0199	0.0221	0.0182	0.0236	0.0307
			NCP	0.3479	6.3044	5.1188	7.5531	4.5026	2.6611
			Power	0.0907	0.7092	0.6189	0.7847	0.5644	0.3714

*L399-: Did you remove the GW associated SNPs before calculating the G1 and G2 matrices? It's not clear if*
*there was one set of G1 and G2 matrices for each of the metabolites or a single pair of G1 and G2 matrices*
*used for every metabolite, but you wouldn't want to include the GW associated SNPs for a metabolite in*
*the G1 and G2 matrices as well. (I see in L148 of the rebuttal that you removed lead SNPs and those within*
*50Kb of the lead SNPs from the matrix, which takes care of my concern. That wasn't clear, however, from*
*the methods. Also, the rebuttal paragraph beginning on L362 discusses three variance component models,*
*which I don't see anywhere in the main text or methods or supplementals. What is this analysis, how was it*
*performed, and where are the results?)*

We highly appreciate the feedback and clarified our methods section with regard to how the $V(G1)$ and
$V(G2)$ matrices were created. The GW associated SNPs were removed prior to the calculation of the $V(G1)$
GRM. Across all analyses the $V(G1)$ and $V(G2)$ GRMs were constant. Depending on whether we analyzed
metabolites classified as “lipids” or “organic acids” we switched between the lipid-specific $V(G3)$ and $V(G4)$
or the organic acid-specific $V(G3)$ and $V(G4)$. We have attempted to clarify this by rewriting the relevant
section in the **Methods** (L456-458). Additional information about the construction of the GRMs has also
been included in **Supplementary Note 2** (on creating metabolite class-specific GRMs). Furthermore, we
realize that **Figure 1** from the previous version of the manuscript was difficult to interpret when
attempting to find which GRM included which SNPs and were used for which variance component. As such
this figure has now been split into **Figure 1** and **Supplementary Figure 1** to separately describe the
curation of the metabolite loci and the construction of the GRMs, and the specification of the four-
variance component models.

The observation of reviewer #3 that the 3-variance component models have not been included in the main
text or supplements is correct. We apologize for any confusion this may have caused. In the first draft of
the manuscript we did not distinguish between metabolite class-specific and non-specific loci in the
estimation of $h^2_{GW-loci}$ (in the second and current versions of the manuscript this is the sum of the class-
specific and non-specific metabolite loci); instead $h^2_{GW-loci}$ was constructed by including all metabolite loci

in a single GRM. Therefore, in the first draft of the manuscript the GREML analyses included 3 GRMs
(hence 3-variance component models), where the second draft and current version include 4 GRMs. The
3-variance component models are no longer relevant in the current version of the manuscript as the
estimates of $h^2_{\text{metabolite-hits}}$ now comprise the sum of $h^2_{\text{class-hits}}$ and $h^2_{\text{notclass-hits}}$ and have therefore been
removed from this and the previous version of the manuscript.

*L406: LDAK estimates of relatedness will be altered by the weights. Using those within a thresholded*
*matrix (the V(G2) estimate) is likely not the same thing as thresholding based on an unadjusted GRM*
*(GCTA-based), or if it is close, you should demonstrate that the same pairs are set to 0 for each approach. It*
*would be simple to compare the two matrices and check to see if the thresholded (set to 0) pairs are the*
*same, and if they are, then that's fine, but if the thresholded pairs set to 0 are different, then you will have*
*to explore how that changes the estimates, particularly since the Zaitlen et al. paper, and all others I'm*
*aware of using close relatives, have used an unweighted GRM for that purpose.*

We thank the reviewer for bringing this issue to our attention. To assess if the same individuals are set to
0 when creating thresholded clumped GRMs in LDAK and GCTA we inspected a scatterplot of the off-
diagonal elements of both GRMs. As can be seen in the scatterplot (**Supplementary Figure 6**) of the LDAK
vs the GCTA thresholded GRMs, there are some discrepancies in the individuals set to zero in one or the
other methods (horizontal/vertical outliers bottom left corner graph). Note that the number of
relationships not classified as unrelated by both methods is very low. Fourteen pairs were classified as
unrelated by the LDAK method, but not by GCTA, and two pairs were classified as unrelated by the GCTA
method, but not by LDAK. Therefore, the effect of using LDAK or GCTA to create the GRM retaining only
the relationships among closely related individuals will be minimal. This comparison has been added to
**Supplementary Note 3.**

**Supplementary Figure 6.** Scatterplot of the off-diagonal elements of the clumped LDAK versus clumped
 GCTA GRMs with the relationships for unrelated individuals (<0.05) set to zero. **(a)** Scatterplot of all off-
 diagonal elements of the clumped LDAK versus clumped GCTA GRMs with the relationships for unrelated
 individuals (<0.05) set to zero. **(b)** . Scatterplot of the <0.05 off-diagonal elements of the clumped LDAK
 versus clumped GCTA GRMs with the relationships for unrelated individuals (<0.05) set to zero.

*L420: How did you compare the models? Supplementary note 3 doesn't describe any formal comparisons.*
 *Log likelihood? AIC? BIC? Something else? As the standard errors are likely to be quite large for the h2SNP*
 *estimate, just finding similar point estimates doesn't seem appropriate or sufficient.*

We would like to thank the reviewer for pointing out the oversight in **Supplementary Note 3** (now
**Supplementary Note 4**) where we compare the Zaitlen model for our metabolomics data with a varying
number of covariates based on the estimate and SE differences between these models with different
numbers of covariates. We had indeed neglected to include formal comparisons when deciding what the
appropriate number of covariates to include should be. We have extracted the Log Likelihood (LogL) for
each of the models for all metabolites across all four platforms, these have been added to **Supplementary**
**Table 19**. We calculated the likelihood ratio test (LRT) to compare the full model with the reduced and
sparse models, these results have been included in **Supplementary Table 19** and described in
**Supplementary Note 4**. The LRT confirms our assessment that on average, for all metabolites on all four
platforms, the sparse model outperforms the full and reduced models.

*L432: What exactly did you resample? How did you perform your random sampling? Why not just use the*
*standard approach for this, the delta method? (Yang 2015 applied that approach, and it is also in the back*
*of the classic Lynch and Walsh "Quantitative Genetics" textbook.)*

We are grateful to the reviewer for pointing out that the description of obtaining the SE's for the
composite variance estimate lacks details. We resampled new variances based on the original means and
variance/covariance matrix of the $V(G1)$, $V(G3)$ and $V(G4)$ GRMs. Random sampling was performed in R by
creating 10,000 multivariate normal distributions (mvrnorm function MASS package) based on the
original means and variance/covariance matrices. On L495-501 we have now provided additional details
about the process by which we obtain the SE's for the composite variance estimates.

We used the delta method in the first version of the manuscript, however, there we observed that the SE's
of the composite variance estimates were driven by the much smaller variances of the $V(G3)$ GRMs, as
opposed to the rather large SE's of the $V(G1)$ GRMs. This resulted in too small SE's for the composite
variance estimates, warranting the alternative approach as applied the second and current versions of the
manuscript.

*Supplementary Note 2: What were the underlying data used for the simulations? Array SNPs, imputed*
*variants, or WGS? That makes a huge amount of difference with these kinds of simulations, because the*
*array and imputed variants are not a random subset of all WGS variants, but rather are better-tagged and*
*are more common than average (and likely more so than unknown causal variants in real data). Using*
*imputed or array data as the underlying simulation leads to a situation where the LD is baked into the*
*simulations already, matching the LDAK model, so of course it will look ‘better’ but simply because of the*
*LD relationships already in the data. The scenario of randomly chosen SNPs is also quite different from your*
*strategy of first clumping variants with plink before making the matrix, because in the real data, you’ve*
*forced all the markers in the matrix to be in low LD with one another. With imputed or array variants used*
*for causal variants in your simulations, the simulations are going to show LD-dependent models working*
*better, though that may not have much relevance to real data where the causal variants are unknown, and*
*likely not array SNPs or well-tagged imputed variants. Additional simulations to establish that thresholding*
*the weighted GRM rather than an unweighted GRM are unbiased would be good to see to address the*
*concern of L406 above.*

The data underlying the simulations in **Supplementary Note 2** (now **supplementary Note 3**) is the same
genotyping data the $V(G1)$ and $V(G2)$ GRMs are based on. This is the combination of all genotyped SNPs
for all the arrays available in the NTR imputed to GONL in order to ensure that all individuals have values
for all genotyped SNPs, regardless of the platform they were genotyped on. We refer to this dataset as the
“cross-imputed genotype set” (see **Methods**). A full comparison of the pros and cons of using LDAK or
GCTA-based GRMs is beyond the scope of the current paper. Similarly, as we don’t use WGS data in the
current paper, a comparison of the current simulations with simulations based on WGS data is beyond the
scope of the current paper. We acknowledge that interpreting the LDAK results as ‘better’ is unjust and
have nuanced our findings in **Supplementary Note 3**. Furthermore, we have added a comparison of the
LDAK and GCTA thresholded GRMs to **Supplementary Note 3**, as described in the response to comment
about L406.

Reviewers' Comments:

Reviewer #3:

Remarks to the Author:

The authors have addressed all of my concerns, and I think the additions and changes are acceptable. The manuscript and the approach are well-done.